



# Characterization of aerosol particles at Cape Verde close to sea and cloud level heights - Part 1: particle number size distribution, cloud condensation nuclei and their origins

Xianda Gong[1], Heike Wex[1], Jens Voigtländer[1], Khanneh Wadinga Fomba[1], Kay Weinhold[1], Manuela van Pinxteren[1], Silvia Henning[1], Thomas Müller[1], Hartmut Herrmann[1], and Frank Stratmann[1]

[1]Leibniz Institute for Tropospheric Research, Leipzig, Germany

**Correspondence:** Xianda Gong (gong@tropos.de)

**Abstract.** In the framework of the MarParCloud (Marine biological production, organic aerosol particles and marine clouds: a Process Chain) project, measurements were carried out on the islands of Cape Verde, to investigate the abundance, properties, and sources of aerosol particles in general and cloud condensation nuclei (CCN) in particular, both close to sea and cloud level heights.

A thorough comparison of particle number concentration (PNC), particle number size distribution (PNSD) and CCN number concentration ($N_{\text{CCN}}$) at the Cape Verde Atmospheric Observatory (CVAO, sea level station) and Monte Verde (MV, cloud level station) reveals that during times without clouds the aerosol at CVAO and MV are similar and the boundary layer is generally well mixed. Therefore, data obtained at CVAO can be used to describe the aerosol particles at cloud level. Cloud events were observed at MV during roughly 58% of the time and during these, a large fraction of particles were activated to cloud droplets.

A trimodal parameterization method was deployed to characterize PNC at CVAO. Based on number concentrations in different aerosol modes, four well separable types of PNSDs were found, which were named the marine type, mixture type, dust type1 and dust type2. Aerosol particles differ depending on their origins. When the air masses came from the Atlantic Ocean, sea spray can be assumed to be one source for particles, besides for new particle formation. For these air masses, PNSDs featured the lowest number concentration in Aitken, accumulation and coarse mode. Particle number concentrations for the sea

spray aerosol (SSA, i.e., the coarse mode for these air masses) accounted for about 3.7% of $N_{\text{CCN,0.30\%}}$ (CCN number concentration at 0.30% supersaturation) and about 1.1% to 4.4% of $N_{\text{total}}$ (total particle number concentration). When the air masses came from the Saharan desert, we observed enhanced Aitken, accumulation and coarse mode particle number concentrations and overall increased $N_{\text{CCN}}$. $N_{\text{CCN,0.30\%}}$ during the strongest observed dust periods is about 2.5 times higher than that during marine periods. However, the particle hygroscopicity parameter $\kappa$ for these two most different periods shows no significant

difference and is generally similar, independent of air mass.

Overall, $\kappa$ averaged 0.28, suggesting the presence of organic material in particles. This is consistent with previous model work and field measurement. There is a slight increase of $\kappa$ with increasing particle size, indicating the addition of soluble, likely inorganic material during cloud processing.



# 1  Introduction

Clouds in the atmosphere are formed when excess water vapor condenses on aerosol particles that serve as cloud condensation nuclei (CCN). Back to 1970s, Twomey (1974) described that an increase in the number of aerosol particles that activate to clouds lead to more but smaller droplets. Albrecht (1989) suggested that smaller droplets then cause suppression in the

formation of precipitation, leading to a prolonged cloud lifetime. Both of these effects enhance the shortwave reflection of clouds, i.e., they lead to a cooling of the atmosphere. In particular, warm low-level clouds located in the boundary layer constitute an important role to the cooling effects due to their abundance and strong cloud albedo effect (Christensen et al., 2016). In recent years, many more aspects of aerosol-cloud interaction were discussed. Considerable progress has been made in understanding the chemical composition and micro-physical properties of aerosol particles that enable them to act as CCN

(Andreae and Rosenfeld, 2008). Particles' ability to act as CCN is largely controlled by aerosol particle size rather than composition (Dusek et al., 2006). However, we still lack understanding of the overall roles of aerosol particles, clouds and their interactions in the climate system, which contribute to the largest uncertainties to estimate the Earth's energy budget (Stocker, 2014).

The mineral dust aerosol in general, Cape Verde mineral dust aerosol in particular, has been studied. Mineral dust from

deserts contributes largely to tropospheric aerosols and impacts air quality of several regions, even of the globe (Ginoux et al., 2001; Huang et al., 2006; Tanaka and Chiba, 2006). The largest dust source is located in the northern hemisphere in the Sahara and Sahel regions (Goudie and Middleton, 2001; Prospero et al., 2002; Ginoux et al., 2012), with millions of tons of mineral dust being transported to Europe and the Middle East, as well as to the Americas yearly (including the Caribbean and the Amazon basin) (Swap et al., 1992; Salvador et al., 2013; Wex et al., 2016). Mineral dust aerosol in the atmosphere can

affect the Earth's radiative budget by directly scattering and absorbing solar and infrared radiation (Goudie and Middleton, 2001; Shao et al., 2011). On the other hand, it can modify cloud properties, i.e., serve as CCN or ice nucleating particles (INPs) (Sassen et al., 2003; DeMott et al., 2003). Karydis et al. (2011) found that the predicted annual average contribution of insoluble mineral dust to CCN number concentration in cloud forming areas is up to 40% on a global basis.

Based on a 3-week field campaign in summer 1973 at Cape Verde, Jaenicke and Schütz (1978) investigated the aerosol

properties such as total size distribution, mass, sea salt, mineral, organic compound content, and found that a total mass of 100 $\mu$g m$^{-3}$ during dust plumes is five times higher than the 20 $\mu$g m$^{-3}$ of clean air masses. Kandler et al. (2011b) also found that the total particle mass concentration during dust plumes was raised by a factor of more than 10 over the maritime mass concentration, demonstrating a strong impact of Saharan dust advection on the aerosol load at Cape Verde. Significant seasonal intrusions of dust from North West Africa affect Cape Verde at surface level from October till March. An hourly PM$_{10}$ value

reached up to 710 $\mu$g m$^{-3}$ at surface level at Cape Verde (Gama et al., 2015). Schladitz et al. (2011b) found that mineral dust particles were mainly in the coarse mode. The variation of the amount of mineral dust is much larger than the variation of the sea salt content in the coarse mode. Also pesticides, polycyclic aromatic hydrocarbons (PAHs), and polychlorinated biphenyl (PCB), all of which originating from the Sahara and Sahel regions, can be incorporated with Saharan dust and then transported to Cape Verde (Garrison et al., 2014).



Considerable studies investigated the marine aerosol in laboratory or in field measurements, and few of them were carried out at Cape Verde or nearby regions. Due to the vast coverage of the Earth's surface by the oceans, wind-driven particle production on the ocean surface is one of the largest global sources of primary atmospheric particle on a mass concentration basis (Warneck, 1999; Modini et al., 2015). Ambient measurements and laboratory studies indicated that the resulting marine aerosol with less than 10 $\mu$m diameter can have a trimodal size distribution, which suggests that several mechanisms are involved in marine aerosol production (Prather et al., 2013; Quinn et al., 2015; Brooks and Thornton, 2018). Marine aerosol number and mass concentrations, chemical composition, and optical and cloud nucleating properties can be changed during transportation, e.g., marine aerosol can carry continental emissions up to thousands of kilometers downwind (Quinn et al., 2015). Marine aerosol impacts Earth's radiation balance by directly scattering solar radiation (Quinn et al., 2017). Besides, ocean physics, biology, and chemistry ultimately control both particle hygroscopicity (Fuentes et al., 2011) and the number of particles that can act as CCN and INPs (Andreae and Rosenfeld, 2008; Wilson et al., 2015; DeMott et al., 2016) in the marine aerosol. On a global basis, marine aerosol makes a contribution of less than 30% to the CCN population (Quinn et al., 2017).

Marine aerosol is the second important aerosol source at Cape Verde when looking at particle mass (Fomba et al., 2014; Salvador et al., 2016). There is always a background of marine aerosol present at Cape Verde (Kandler et al., 2011a). Based on a 5-year measurement at Cape Verde, Fomba et al. (2014) found that the mean mass concentration of sea salt was 11.00$\pm$5.10 $\mu$g m$^{-3}$ (corresponding to total mass of 47.20$\pm$55.50 $\mu$g m$^{-3}$). Additionally during summer, elevated concentrations of organic material were observed to originate from marine emissions. A summer maximum was observed for non-sea-salt sulfate and was connected to periods when air mass inflow was predominantly of marine origin, indicating that marine biogenic emissions were a significant source. Schladitz et al. (2011b) found that the Aitken and accumulation mode particles were mainly composed of the marine aerosol, whereas coarse mode particles were composed of sea salt and a variable fraction of Saharan mineral dust.

As outlined above, Saharan dust and sea salt dominate PM$_{10}$ particle composition (more than 70%) near the surface at the Cape Verde (Fomba et al., 2014; Salvador et al., 2016). In addition, Cape Verde is rich in other kinds of aerosols from both continental and marine sources. Biomass burning aerosols produced from October to November in sub-sahelian latitudes had a clear influence on the content of elemental carbon (EC) recorded at Cape Verde, but a small impact on PM$_{10}$ (Salvador et al., 2016), as particles originating from biomass burning layer usually stay at high altitude (1500 - 5000m) (Tesche et al., 2009; Heinold et al., 2011; Lieke et al., 2011).

Overall, there are diverse sources of less or more hygroscopic particles which might contribute to aerosols at Cape Verde. Pringle et al. (2010) used an atmospheric chemistry model to simulate global fields of the effective hygroscopicity parameter, represent as $\kappa$ (Petters and Kreidenweis, 2007), which roughly describes the influence of chemical composition on CCN activity of aerosol particles. An annual cycle of monthly mean $\kappa$ value at the surface of the Cape Verde was reported in Pringle et al. (2010). To the best of our knowledge, the only filed measurement of particle hygroscopicity at Cape Verde was carried out by Schladitz et al. (2011a). Here, these model results and field measurement values will be compared with those obtained from in-situ measurements during our measurement campaign in the framework of the MarParCloud (Marine biological production, organic aerosol particles and marine clouds: a Process Chain) project.





Atmospheric boundary layer (ABL) is the region in the lowest part of troposphere (below 1000 m above the ground), where the Earth's surface strongly influences temperature, moisture, and wind through the turbulent transfer of air mass. Most particles are emitted or formed in the ABL with temporally varying sources (Rosati et al., 2016b). Extensive data sets from ground-based aerosol properties studies are available. One major point of interest is to know whether ground-based measurements can be

used to infer aerosol properties at cloud level. Previous field measurements at Po Valley and Netherlands found that during the development of a newly mixed layer, the estimation of altitude-specific data from surface measurements may be problematic (Rosati et al., 2016b, a). Once the ABL was fully mixed, a constant extinction coefficients (Rosati et al., 2016b) and particle hygroscopicity (Rosati et al., 2016a) were observed at all altitudes within ABL. Wex et al. (2016) found for the marine aerosol on Barbados, that the particle number size distribution (PNSD) on ground and throughout the sub-cloud level showed good

agreement.

During the MarParCloud project, we set two measurement stations, one close to the sea level (10 m a.s.l) and one on a mountaintop (744 m a.s.l), to characterize aerosols properties, including particle number concentration (PNC), PNSD and CCN number concentration ($N_{CCN}$). In addition, a kite and balloon borne (Helikite) measurement was carried out to characterize vertical profiles of meteorological parameters at Cape Verde. This offered a unique opportunity to compare particle properties

close to the sea level and higher up in the marine boundary layer (MBL) height.

In a series of two papers, we aim to provide a quantitative understanding regarding the abundance, properties and source of aerosol particles in general, CCN and INPs in particular close to both sea and cloud level heights. In this paper, we will (1) compare aerosol properties measured close to see level and at a mountaintop to examine the representativeness of ground based measurements to the MBL and (2) present a thorough characterization of CCN with respect to their hygroscopicity and number

concentrations for different air masses. To the best of our knowledge, both of this will be presented here for the Cape Verde for the first time. In a companion paper, we will examine the abundance and properties of INPs from several different sources, namely sea surface microlayer and under layer water from the ocean, airborne close to sea and cloud level, and cloud water of warm cloud. This study is the first in a series of publications to come from the MarParCloud project. For more information about the campaign itself and a more detailed analysis of the meteorological situation, we refer to an upcoming overview

paper (in preparation by van Pinxteren et al.,), which will also cover a thorough size-resolved chemical composition analysis of particles close to the sea level and on the mountaintop.

## 2   Experiment and methods

### 2.1   Sampling sites and campaign setup

The measurements were carried out on São Vicente island in Cape Verde from 13 September to 13 October, 2017. Located

in the Atlantic Ocean, São Vicente island is ∼900 km off the African coast. The region experiences constant northeasterly winds. The average annual temperature at Cape Verde is 23.6 ± 4.0 °C (mean±1 standard deviation). It is an arid region with a maximum of 24-350 mm rainfall per year. The precipitation frequency is about 3 to 10 events annually, mainly between August and October. More details of the meteorological conditions at Cape Verde can be found in Carpenter et al. (2010).





Three measurement stations were set up at Cape Verde, i.e., Cape Verde Atmospheric Observatory (CVAO), Monte Verde station (MV) and Ocean Station (OS, will be discussed only in the companion paper). CVAO (16°51′49 N, 24°52′02 W) is located at the northwestern shore of the São Vicente island, 70 m from the coastline at about 10 m a.s.l. An aerosol $PM_{10}$ inlet, employed to remove particles larger than 10 $\mu$m in aerodynamic diameter, was installed on top of a 32 m tower. Downstream

of the aerosol inlet there was a vertical stainless steel sampling pipe (32 m long, 1/2 inch outer diameter), installed together with a diffusion dryer which was placed directly on top of the measurement container. Aerosol entered the inlet on top of the mast and was transported through the tube and the dryer. Downstream of the dryer and inside of the container, the aerosol was split isokinetically and distributed to various instruments, including a Mobility Particle Sizer Spectrometer (MPSS), an Aerodynamic Particle Sizer (APS) and a Cloud Condensation Nuclei counter (CCNc). Besides, airborne measurements were

carried out at the CVAO using a Helikite, to characterize the vertical profiles of temperature, relative humility (RH), wind speed and direction.

MV (16°52′11 N, 24°56′02 W) is located on the top of Monte Verde (744 m a.s.l), ~7 km away to the west of the CVAO. An aerosol inlet was installed on the roof of a building which overall had a cut-off size of 4 $\mu$m. A vertical stainless steel sampling pipe (2 m long, 1 inch diameter) and a diffusion dryer were placed downstream of the aerosol inlet. Downstream of the dryer

and inside the building, the sample aerosol was split isokinetically to MPSS and CCNc. An overview of the sampling site and instruments can be seen in Tab. 1. In the following, we will briefly introduce the different measurement techniques applied in this study, including calibrations, measurements and data processing.

## 2.2 Balloon measurement

Vertical profile of meteorological parameters was taken at CVAO. The measurements were achieved using a 16 m³ Helikite

(Allsopp Helikites Ltd, Hampshire, UK), a unique combination of a tethered balloon and a kite. Helikites are designed to be operated under extreme weather conditions. The kite was attached to a 3 mm Dyneema rope (2000 m long, ~4.6 g m$^{-1}$, Lyros D-Pro 3 mm, breaking load 950 daN, working elongation <1%) and operated by a winch. Under calm condition, the Helikite has a net load capacity of ~8 kg. Under windy condition, the pull increases significantly and the net load capacity reaches about 16 kg at 6 m s$^{-1}$. Depending on the prevailing conditions, meteorological measurements of up to an altitude of about

1200 m could be carried out. The measuring system, built by Leibniz Institute for Tropospheric Research (TROPOS), was attached to the rope 20 m below the helikite. All sensors were selected and tested individually in the laboratory at TROPOS. Wind speed was measured using a differential pressure sensor together with a pitot tube, wind direction was determined from an orientation sensor (compass) of a wind vane. Data was recorded with a measuring frequency of 2 Hz, stored in a SD card and additionally transmitted to a ground station (via XBee). Our aim was to characterize the atmospheric boundary layer in

terms of mixing state, which can provide insights to questions regarding the connection between ground-based measurements and the free troposphere.





**Table 1.** Measured and derived parameters and the respective instrumentation used at CVAO and MV.

| Measurement site | Location | Parameter | Abbreviation | Instrument | Measurement range |
|---|---|---|---|---|---|
| CVAO | 16°51′49 N, 24°52′02 W | | | | |
| | Inlet height 42 m a.s.l | | | | |
| | | Particle number size distribution | PNSD | MPSS and APS system | 10 to 10000 nm |
| | | Particle number concentration | $N_{total}$ | intergrated PNSD | - |
| | | CCN number concentration | $N_{CCN}$ | CCNc | - |
| | | Particle hygroscopicity | $\kappa$ | CCNc with | SS=0.15%, 0.20%, 0.30%, |
| | | | | MPSS and APS system | 0.50% and 0.70% |
| | | Vertical profile of temperature and RH | - | Balloon | height up to 1200 m |
| MV | 16°52′11 N, 24°56′02 W | | | | |
| | Inlet height 746 m a.s.l | | | | |
| | | Particle number size distribution | PNSD | MPSS system | 10 to 850 nm |
| | | Particle number concentration | $N_{total}$ | intergrated PNSD | - |
| | | CCN number concentration | $N_{CCN}$ | CCNc | - |
| | | Particle hygroscopicity | $\kappa$ | CCNc with | SS=0.15%, 0.21%, 0.33%, |
| | | | | MPSS system | 0.56% and 0.79% |



### 2.3 Particle number size distribution

PNSDs were measured in the size range from 10 nm to 10 $\mu$m using a TROPOS-type MPSS (Wiedensohler et al., 2012), and an APS (Aerodynamic Particle Sizer model 3321, TSI Inc., St. Paul, MN, USA). The APS data was accounted for the multiple charge correction of MPSS data in the inversion of measured PNSD (Wiedensohler, 1988; Pfeifer et al., 2016). The

combined PNSD is then given on the base of the volume equivalent particle diameter. More details about the combined MPSS and APS PNSDs S can be found in Schladitz et al. (2011b). Size-dependent particle losses caused by diffusion, deposition and sedimentation within the inlet were corrected for, by utilizing the empirical particle loss calculator (von der Weiden et al., 2009). The size dependent particle losses are shown in the supplement, Fig. S1. Total particle number concentrations ($N_{\text{total}}$) were calculated from the measured PNSDs accounting for the size-dependent particle losses. The MPSS and APS were calibrated

before, during and after the intensive field study. More details about calibration methods can be found in Wiedensohler et al. (2018).

### 2.4 Cloud condensation nuclei

$N_{\text{CCN}}$ was measured using a Cloud Condensation Nuclei counter (CCNc, Droplet Measurement Technologies, Boulder, USA, Roberts and Nenes, 2005). The CCNc is a cylindrical continuous-flow thermal-gradient diffusion chamber, establishing a

constant streamwise temperature gradient to adjust a quasi constant centerline supersaturation. The sampled aerosol particles are guided within a sheath flow through this chamber and can become activated to droplets, depending on the supersaturation and the particles' ability to act as CCN.

During our study, the supersaturation was varied between ~0.15 % to ~0.79 % at a constant total flow rate of 0.5 L min$^{-1}$. To assure stable column temperature, the first 5 minutes and the last 30 seconds of each 12-minute long measurement at each

supersaturation were excluded from the data analysis. The remaining data points were averaged. A supersaturation calibration (following protocol by Gysel and Stratmann, 2013) was done at the cloud laboratory of the TROPOS prior to and after the measurement campaign in order to determine the relationship between the temperature gradient along the column and the effective supersaturation. Calibrated supersaturation set-points were 0.15 %, 0.20 %, 0.30 %, 0.50 % and 0.70 % of CVAO CCNc and 0.15 %, 0.21 %, 0.33 %, 0.56 %, 0.79 % of MV CCNc. These values were used for further calculations.

According to Köhler theory (Köhler, 1936), whether or not a particle can act as a CCN depends on its dry size, chemical composition and the maximum supersaturation it encounters. Petters and Kreidenweis (2007) presented a method to describe the relationship between particle dry diameter and CCN activity using the hygroscopicity parameter $\kappa$. $\kappa$ values reported in this study were calculated as follows, assuming the surface tension of the examined solution droplets $\sigma_{s/\alpha}$ to be that of pure water:

$$\kappa = \frac{4A^3}{27d_{\text{crit}}^3 \ln^2 S} \tag{1}$$

with

$$A = \frac{4\sigma_{s/\alpha}M_\omega}{RT\rho_\omega} \tag{2}$$





where $d_{\mathrm{crit}}$ is the critical diameter above which all particles activate into cloud droplets for a given supersaturation. $S$ is the supersaturation. $M_\omega$ and $\rho_\omega$ are the molar mass and density of water, while $R$ and $T$ are the ideal gas constant and the absolute temperature, respectively. To derive $d_{\mathrm{crit}}$, simultaneously measured $N_{\mathrm{CCN}}$ and PNSD are used. Thereto, it is assumed that all particles in the neighborhood of a given particle diameter have a similar $\kappa$, meaning that the aerosol particles are

internally mixed. At a given supersaturation, a particle can be activated to a droplet once its dry size is equal to or larger than $d_{\mathrm{crit}}$. Therefore, $d_{\mathrm{crit}}$ is the diameter at which $N_{\mathrm{CCN}}$ is equal to the value of the cumulative particle number concentration, determined via integration from the upper towards the lower end of the PNSD. Values for $\kappa$ can be calculated with $d_{\mathrm{crit}}$ and the corresponding supersaturation, based on Eq. 1. The inferred $\kappa$ values correspond to particles with sizes of roughly $d_{\mathrm{crit}}$. The uncertainty in $\kappa$, which results from uncertainties of the PNSD measurements and the supersaturations of the CCNc, was

determined by applying a Monte Carlo simulation (MCS) in a similar fashion as done by Kristensen et al. (2016) and Herenz et al. (2018). A detailed description of this method is provided in the supplement. Note that the particle losses inside the CCNc (discussed in Rose et al., 2008) were also considered before $\kappa$ was calculated.

## 3   Results and discussion

### 3.1   Overview of the meteorology

Time series of meteorology parameters, including the wind speed, temperature and RH at CVAO and MV, as well as wind direction at CVAO are shown in Fig. 1. The wind speed varied from 0.6 to 9.7 m s$^{-1}$, with a mean of 6.0 m s$^{-1}$ at CVAO. The variation of wind speed at MV is similar to that at CVAO. Fig. 2 shows the wind rose plot based on hourly average of wind speed and direction at CVAO. Clearly, the CVAO station experienced constant northeasterly winds during this campaign. The temperature and RH were measured by digital temperature humidity sensor (Davis Instruments, 7346.070). The accuracy of

the temperature sensor is $\pm\,0.3\,^\circ$C and the humidity sensor is $\pm\,2\%$ below 90% and $\pm\,4\%$ above 90%. The temperature and RH at CVAO varied from 25.6 to 28.3 $^\circ$C and 70.0% to 90.5%, with a mean of 26.6 $^\circ$C and 81.0 %, respectively. Obviously, temperature at MV was lower than that at CVAO, ranging from 18.9 to 25.4 $^\circ$C, with a mean of 21.2 $^\circ$C. The measured RH was 100% during more than half of the campaign at MV. Note that due to the instrument detection limit, RH=100% is not accurate. However, the RH=100% indicated that the MV station was often in a cap cloud. More precise determination of cloud

events and influences of cloud on aerosol particles will be discussed in section 3.3. Note that all the times presented here are in UTC (corresponding to local time+1). For better comparison, all PNC and $N_{\mathrm{CCN}}$ reported in this study are given for standard temperature and pressure (STP, 0 $^\circ$C and 1013.25 hPa).

During the MarParCloud campaign, 19 vertical profiles on 10 different days were taken. Profiles of up to about 1200 m could be measured. The inversion layer heights were determined by the measurements. The MBL was typically well mixed

with boundary layer heights between about 550 and 1100 m, as shown by blue rectangles in Fig. 1. It is indicated that there were 3 cases of a decoupled boundary layer during the whole campaign, as shown by red dots in Fig. 1. Therefore, to be sure to represent aerosol collected at Cape Verde, we used backward trajectories starting at 200 m altitude to represent MBL air mass origins in this study. Exemplary data from one balloon measurement can be found in the supplement (Sec. S3).





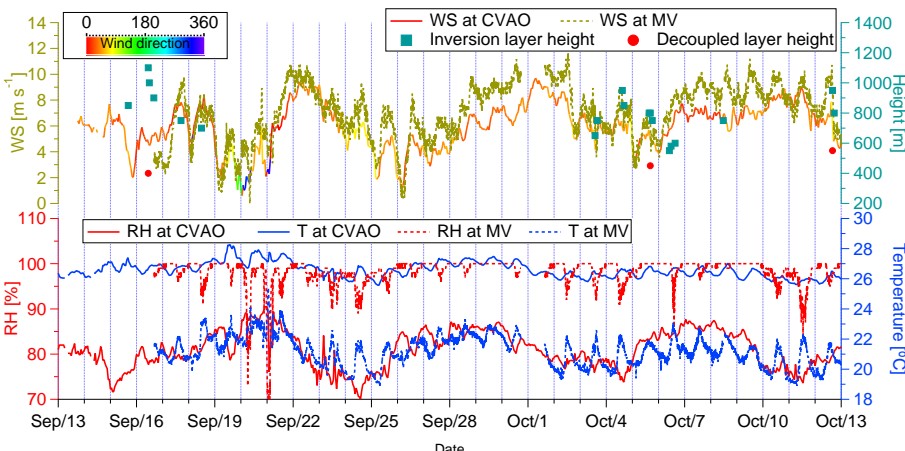

**Figure 1.** Time series of wind speed, wind direction (CVAO only), RH and temperature. Parameters measured at CVAO are shown in solid lines and at MV in dashed lines. Time series of inversion layer height in blue squares and decoupled layer height in red dots.

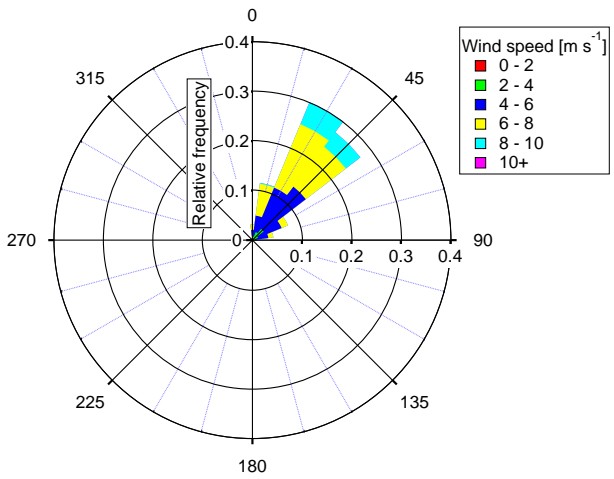

**Figure 2.** Wind rose based on hourly averages of wind speed and direction (measured at CVAO).

## 3.2 Particle characterization

This section will first discuss PNSDs and PNC at CVAO. A trimodal log-normal parameterization method is adopted to characterize the temporal variation of PNC in three modes. It is used to classify the particles into different types based on PNC in different modes. Lastly, to get insights into possible particle sources, we studied the air mass origin and transport through

5   backward trajectory analysis. Calculations were performed with the HYSPLIT (HYbrid Single-Particle Lagrangian Integrated Trajectory) Model (Stein et al., 2015; Rolph, 2003).





### 3.2.1 Particle number size distribution and concentration

Particle size is one of the most important parameters to characterize the behavior of aerosols. Fig. 3 presents contour plots for PNSDs of super-micron particles in the upper panel and submicron particles in the lower panel. In order to show details of super-micron particles, we adopted different color bar scales for submicron and super-micron particles. Most of the time, two submicron modes (Aitken and accumulation mode) and one super-micron mode (coarse mode) are observed. The Aitken mode is observed from ~10 to ~80 nm and accumulation mode is observed from ~80 to ~1000 nm. However, from 03:30 to 20:00 21 September and from 09:30 28 to 18:30 30 September, the submicron particles only exhibited a unimodal distribution. The super-micron particles exhibited high concentration during those periods. $N_{total}$ was changed significantly, from ~200 to ~1500 cm$^{-3}$ with a median of ~700 cm$^{-3}$, shown as black line in the lower panel of Fig. 3.

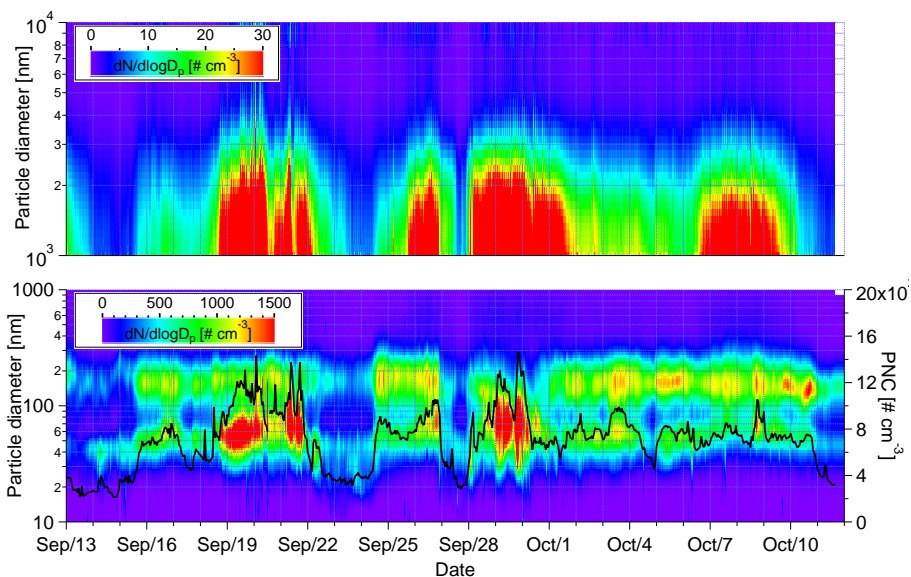

**Figure 3.** Contour plots for PNSDs of 1000 nm to 10 $\mu$m in the upper panel and 10 to 1000 nm in the lower panel. The color scale indicates dN/dlogDp in cm$^{-3}$. Time series of $N_{total}$ is shown in black line in the lower panel.

Particles of different sizes have different formation routes, sources and behaviors. To better define the modes of our data, a trimodal log-normal parameterization method is adopted. The log-normal distribution was expressed by Seinfeld and Pandis (2016) and the parameterization function is as follows:

$$\frac{dN}{dlogD_p} = \sum_{i=1}^{n} \frac{N_i}{\sqrt{2\pi}\log_{10}\sigma_i} \exp\left(-\frac{(\log_{10}D_p - \log_{10}D_i)^2}{2(\log_{10}\sigma_i)^2}\right) \tag{3}$$

where $N_i$ is the total number concentration of the $i$ mode; $D_i$ is the geometric mean diameter of the $i$ mode; $\sigma_i$ is the geometric standard deviation of the number $i$ mode distribution. Every PNSD was individually parametrized by a trimodal distribution, where the number of $i = 1, 2, 3$ stand for Aitken, accumulation and coarse mode, receptively. For each PNSD, we sought for an optimal fitting function, until the coefficient of determination (R$^2$) was larger than 0.97.





Time series of PNC in Aitken mode ($N_{\text{Aitken}}$), accumulation mode ($N_{\text{accumulation}}$), and coarse mode ($N_{\text{coarse}}$), together with sum of $N_i$ and measured $N_{\text{total}}$ are shown in Fig. 4. Due to the unimodal distribution of submicron particles from 03:30 to 20:00 21 and from 09:30 28 to 18:30 30 September, the trimodal log-normal fitting did not work properly, so we did a bimodal log-normal fitting instead, with one submicron mode ($N_{\text{submicron}}$, as shown by purple dots) and one coarse mode.

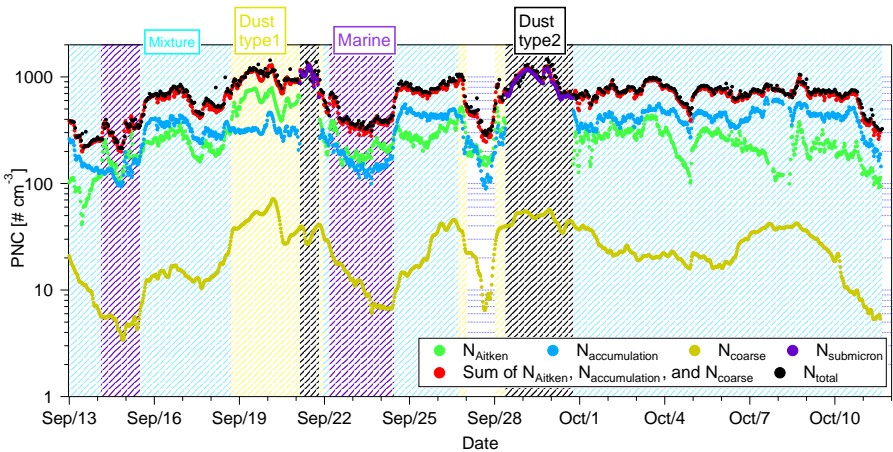

**Figure 4.** Time series of $N_{\text{Aitken}}$, $N_{\text{accumulation}}$, $N_{\text{coarse}}$, $N_{\text{submicron}}$, sum of $N_i$ and $N_{\text{total}}$ at CVAO. The different shadow colors indicate different aerosol type periods.

5      PNC showed large variability during our measurement. $N_{\text{Aitken}}$ and $N_{\text{accumulation}}$ varied from 41 to 789 and 89 to 639 cm$^{-3}$, with a median of 244 and 354 cm$^{-3}$, respectively. Generally, Aitken mode particles are produced by homogeneous and heterogeneous nucleation processes, formed during natural gas-to particle condensation. Accumulation mode particles are formed mainly by coagulation of smaller particles or condensation of vapors onto existing particles, during which they grow into that size range (Seinfeld and Pandis, 2016). Therefore, when $N_{\text{accumulation}}$ is higher than $N_{\text{Aitken}}$, this indicates long-range transport and a more aged aerosol. $N_{\text{coarse}}$ varied from 3 to 71 cm$^{-3}$, with a median of 21 cm$^{-3}$. Coarse mode particles are mostly emitted to the atmosphere from natural sources, e.g., marine aerosol, mineral dust or biological materials.

### 3.2.2 Particle classification and origins

To better understand the particle sources and features, we divided the data from the campaign into different periods. An overview of the classification criteria and features of the different resulting aerosol types are summarized in Tab. 2. Details about the classification criteria are discussed in the supplement. Classification results are shown as different background color in Fig. 4. Note that from 00:00:00 27 to 00:00:00 28 September, $N_{\text{total}}$ suddenly decreased. This might due to the wet deposition that happened before the air masses arrived at the measurement site. The precipitation is an output parameter of the calculated NOAA HYSPLIT backward trajectories. From 00:00:00 27 to 00:00:00 28 September, the total precipitation (sum of precipitation of 144 segment endpoints) exceeded a value of 7 mm in the past 144 hours (corresponding to 6 days) of each backward trajectory history. Therefore, this period was not included in the aerosol classification.



**Table 2.** Classification criteria and features of 4 different particle types

| Aerosol type | Criteria | $N_1$ (cm$^{-3}$) median±std* | $N_2$ (cm$^{-3}$) median±std | $N_3$ (cm$^{-3}$) median±std | Feathers $N_{total}$ (cm$^{-3}$) median±std | $N_{CCN, 0.30\%}$ (cm$^{-3}$) median±std | Shape of PNSD |
|---|---|---|---|---|---|---|---|
| Marine | $N_{Aitken} > N_{accumulation}$ $N_{coarse} < 25$ cm$^{-3}$ | 189±58 | 143±41 | 7±6 | 369±124 | 190±49 | visible Hoppel minimum at 70 nm |
| Mixture | $N_{Aitken} < N_{accumulation}$ | 247±78 | 405±102 | 20±10 | 725±173 | 478±76 | visible Hoppel minimum at 80 nm |
| Dust type1 | $N_{Aitken} > N_{accumulation}$ $N_{coarse} > 25$ cm$^{-3}$ | 556±134 | 312±50 | 39±11 | 952±173 | 332±44 | visible Hoppel minimum at 100 nm |
| Dust type2 | Single mode in submicron $N_{coarse} > 25$ cm$^{-3}$ | - | - | 44±8 | 994±218 | 503±105 | no visible Hoppel minimum |

* one standard deviation





Fig. 5 shows the median of PNSDs of the 4 different aerosol types, with a linear (top) and a logarithmic (bottom) scaling on the y axis. The error bar indicates the range between 25% and 75% percentiles. PNSDs which have $N_{Aitken}$ larger than $N_{accumulation}$ and $N_{coarse}$<25 cm$^{-3}$ are attributed to the "marine type" in this work. PNSDs resembling those show three modes, i.e., Aitken, accumulation and coarse mode, which can be clearly distinguished, as shown in blue line in Fig. 5. The marine type featured the lowest $N_{Aitken}$, $N_{accumulation}$ and $N_{coarse}$ values of 189, 143 and 7 (median) cm$^{-3}$, respectively. The minimum between the Aitken and accumulation mode of PNSDs (Hoppel minimum; see Hoppel et al., 1986) at roughly 70 nm indicates the sizes above which particles had previously been activated to cloud droplets during the history of the air mass at least once. When passing through a cloud, soluble material is added to the activated particles by aqueous-phase chemistry, increasing particulate mass and hence also the size of those particles. The coarse mode particles can be also assumed to be sea spray aerosol (SSA) during the marine type period, as discussed in previous studies (Modini et al., 2015; Wex et al., 2016). A decent correlation ($R^2$=0.69, p<0.01) was found between SSA number concentration and wind speed (supplement, Fig. S6). Modini et al. (2015) also observed that SSA number concentration correlated with local wind speed, which is consistent with the fact that SSA are generated from the process associated with the agitation of the sea surface by air moving above it. The SSA accounted for 1.1% to 4.4% of $N_{total}$ at CVAO (wind speed from 4 to 10 m s$^{-1}$), which is relatively low comparing to e.g., Wex et al. (2016) who found the SSA particles contributed to 4% to 10% of $N_{total}$ (wind speed up to 14 m s$^{-1}$) for the marine aerosol on Barbados. Fig. 6 shows the 6-day backward trajectories with 1 hour time resolution ending at 200 m above the CVAO. Looking at Fig. 6(a), which displays the marine periods, the backward trajectories clearly featured paths over the Atlantic Ocean and traveled to Cape Verde. Non of the backward trajectories touched the European or African continent.

PNSDs that have a larger $N_{accumulation}$ than $N_{Aitken}$ are attributed to the "mixture type" in this work, shown as green line in Fig. 5, with three modes, i.e., Aitken, accumulation and coarse mode, which can be clearly distinguished. $N_{Aitken}$, $N_{accumulation}$ and $N_{coarse}$ have a value of 247, 405 and 20 (median) cm$^{-3}$, respectively. The Hoppel minimum of the mixture type is at roughly 80 nm. The respective backward trajectories, colored in green in Fig. 6(b), showed that the related air mass came from the north Atlantic Ocean and spend some days over southern Europe and northern Africa. Anthropogenic aerosol and mineral dust may be incorporated into air parcels, and transported to Cape Verde, causing higher value of Aitken, accumulation and coarse mode particles than in the marine type.

PNSDs with larger $N_{Aitken}$ than $N_{accumulation}$ and $N_{coarse}$>25 cm$^{-3}$ are attributed to the "dust type1" in this work, shown as red line in Fig. 5. PNSDs attributed to those show three modes, i.e., Aitken, accumulation and coarse mode, which can be clearly distinguished. $N_{Aitken}$, $N_{accumulation}$ and $N_{coarse}$ had a value of 556, 312 and 39 (median) cm$^{-3}$, respectively. The Hoppel minimum of the mixture type is at roughly 100 nm. The respective backward trajectories, colored in red in Fig. 6(c), featured two pathways. One group of air mass originated from the north Atlantic Ocean and stayed few days over southern Europe and northern Africa. One group of air mass came from the Saharan desert.

It is interesting to note that the Hoppel minimum is at the lowest diameter for the marine air mass (~70 nm), compared to all other air masses. This suggests that the supersaturation in the clouds forming in the clean marine air masses is highest, as there is less surface area for the water vapor to condense onto during cloud formation.





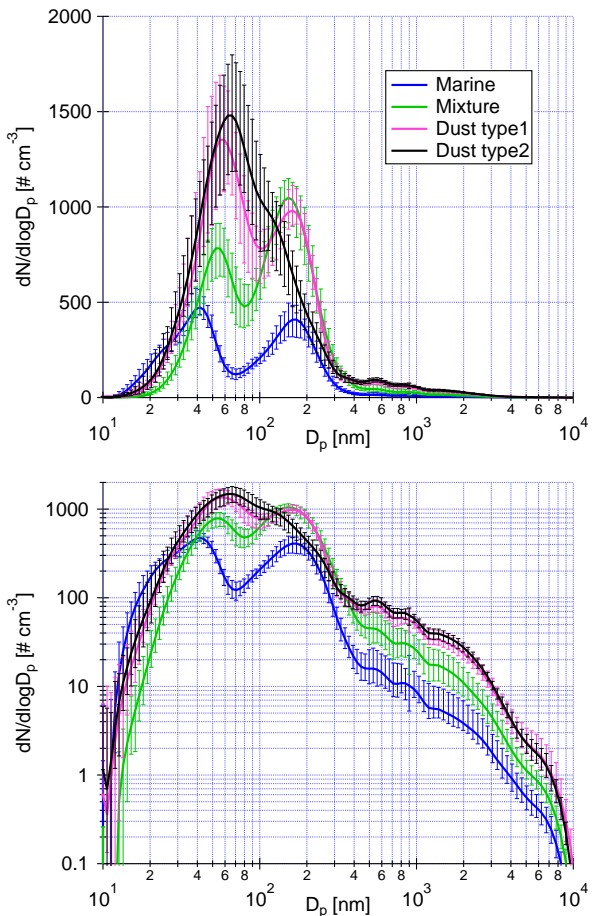

**Figure 5.** The median of PNSDs of marine type (blue), mixture type (green), dust type1 (purple) and dust type2 (black), with a linear (top) and a logarithmic (bottom) scaling on the y axis. The error bar indicates the range between 25% and 75% percentiles.

PNSDs which featured a single mode in the submicron size range are attributed to "dust type2", shown as black line in Fig. 5. No visible Hoppel minimum can be seen. The dust type2 featured highest $N_{total}$ and $N_{coarse}$ values of 994 and 44 (median) cm$^{-3}$, respectively. It is worth to be mentioned that previous field measurement at the Saharan desert found similar PNSDs to what we observed in this study (Kaaden et al., 2009; Kandler et al., 2009; Weinzierl et al., 2009). We assumed
5   dust type2 is the heaviest dust plume period during this campaign. The respective backward trajectories, colored in black in Fig. 6(d), showed that related air masses originated from the Saharan desert.

The higher $N_{coarse}$ during dust type1 and type2 is due to the direct dust aerosol from the Saharan desert. Schladitz et al. (2011b) also found that the higher coarse mode number concentration at Cape Verde originated from the Saharan desert. Besides, a very high concentration of Aitken mode particles was observed during dust type1 and dust type2 periods. A previous
10  study also found that an African-influenced period showed a great enhancement in the Aitken mode particles and an overall





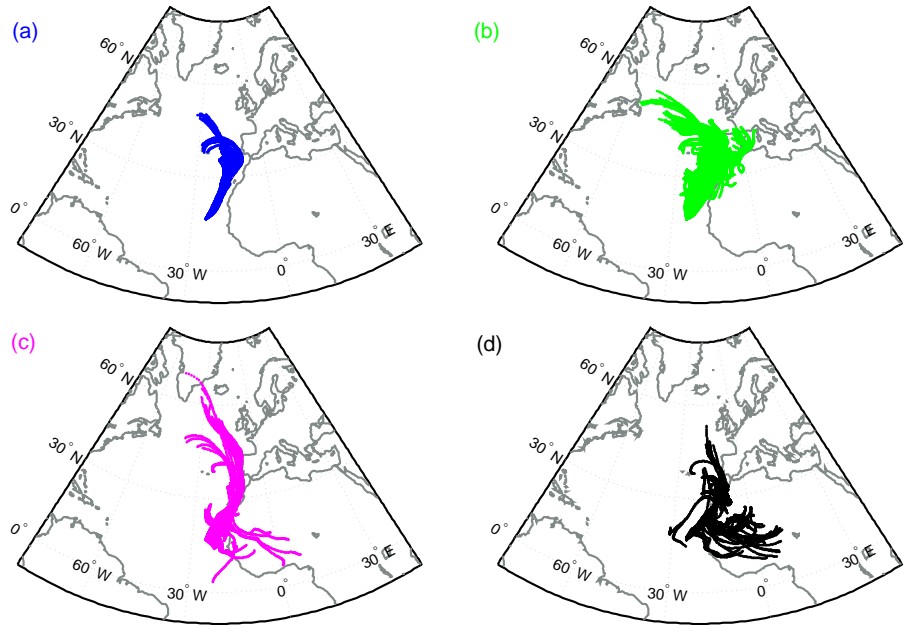

**Figure 6.** 6-day backward trajectories arriving at CVAO at a height of 200 m with 1 h resolution for marine type (a), mixture type (b), dust type1 (c) and dust type2 (d). Each calculation is shown as a separate dot, which is separately visible when air masses moved fast.

increase in the number of particles of all sizes (Allan et al., 2009). Nie et al. (2014) found that new particle formation and growth happened in the remote ambient atmosphere during the strongest observed dust episodes. Both the formation and growth rates of particles in the diameter range of 15–50 nm were enhanced during the dust episodes. Therefore, there are different factors contributing to the observed high $N_{Aitken}$ and $N_{accumulation}$ during dust plumes, such as direct transport of particles from the
5   desert and Sahel region and additional new particle formation and growth.

In short summary, in section 3.2, based on number concentrations in different aerosol modes, an aerosol classification was done, and four well separable types of PNSDs were found, i.e., the marine type, mixture type, dust type1 and dust type2. Marine type particles are mainly from the Atlantic Ocean, while dust type particles are mainly from the Saharan desert. Mixture type particles are a combination of marine, anthropogenic and dust particles. Backward trajectories support this classification and
10  analysis.

## 3.3   Comparison of CVAO and MV

In this section, we will compare the PNC, PNSDs and $N_{CCN}$ at CVAO and MV. Cloud events are identified based on the difference of integrated PNC between MV and CVAO. Cloud effects on PNSDs and $N_{CCN}$ will also be discussed.



### 3.3.1 Comparison of PNC and PNSD

PNSDs from 10 to 800 nm were measured by MPSS and a bimodal log-normal parameterization was adopted to calculate $N_{\text{Aitken}}$ and $N_{\text{accumulation}}$ at MV. Fig. 7 shows the time series of PNC in the size range between 10 and 800 nm at CVAO ($N_{\text{10-800nm}}^{\text{CVAO}}$) and MV ($N_{\text{10-800nm}}^{\text{MV}}$) in the upper panel, PNC of accumulation mode at CVAO ($N_{\text{accumulation}}^{\text{CVAO}}$) and MV ($N_{\text{accumulation}}^{\text{MV}}$)

in the middle panel and PNC of Aitken mode at CVAO ($N_{\text{Aitken}}^{\text{CVAO}}$) and MV ($N_{\text{Aitken}}^{\text{MV}}$) in the lower panel. The variation of $N_{\text{10-800nm}}^{\text{CVAO}}$ and $N_{\text{10-800nm}}^{\text{MV}}$ were similar sometimes, e.g., from 23 to 25 September. However, sometimes the concentrations at MV were obviously lower than the respective values at CVAO, at least for $N_{\text{10-800nm}}^{\text{MV}}$ and $N_{\text{accumulation}}^{\text{MV}}$, as e.g. from 5 to 9 October. Such a decrease was sometimes, but not always, also observed for $N_{\text{Aitken}}^{\text{MV}}$. This is a typical observation for cloudy air, in which particles from the accumulation mode and maybe also some from the Aitken mode are activated to cloud droplets which are

then removed in the aerosol inlet on MV. When the ratio of $N_{\text{accumulation}}^{\text{MV}}$ to $N_{\text{accumulation}}^{\text{CVAO}}$ was lower than 0.85, we assumed that MV is in the cloud. When the trimodal fitting function did not work for the CVAO data set (from 03:30 to 20:00 21 and from 09:30 28 to 18:30 30 September), a slightly different approach was needed. For that, we used the ratio of PNC in the size range between 80 and 800 nm at MV ($N_{\text{80-800nm}}^{\text{MV}}$) to that at CVAO ($N_{\text{80-800nm}}^{\text{CVAO}}$) (replacing the ratio of $N_{\text{accumulation}}^{\text{MV}}$ to $N_{\text{accumulation}}^{\text{CVAO}}$). When this ratio was lower than 0.75, we assumed that MV is in the cloud. It is described in more detail in the supplement

how this ratio was derived separately for cases with trimodal and bimodal fitting. The time for cloud events is shown as red shadows in Fig. 7. As outlined above in the meteorology part, we observed RH=100% at MV. Fig. S8 shows the time series of RH at MV together with the time for cloud events as red shadows. It is clear that times with RH=100% are consistent with cloud events identified as described above, which verifies our identification of cloud events.

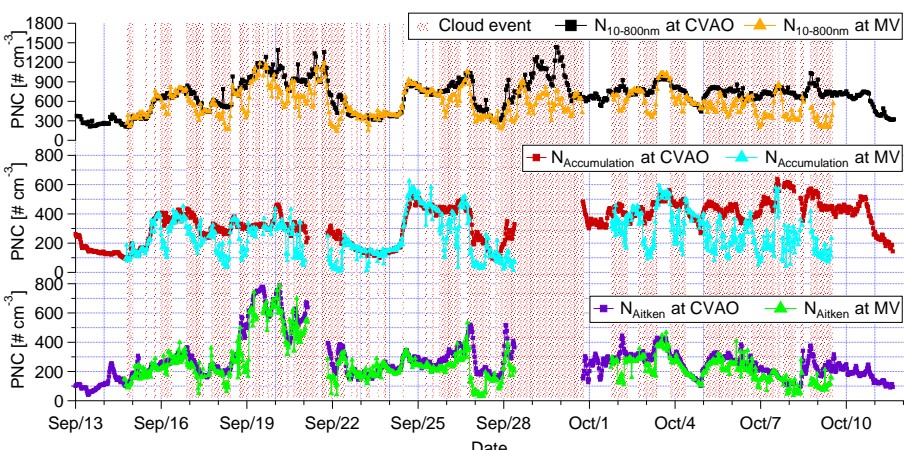

**Figure 7.** Time series of $N_{\text{10-800nm}}^{\text{CVAO}}$ and $N_{\text{10-800nm}}^{\text{MV}}$ in the upper panel, $N_{\text{Aitken}}^{\text{CVAO}}$ and $N_{\text{Aitken}}^{\text{MV}}$ in the middle panel, and $N_{\text{accumulation}}^{\text{CVAO}}$ and $N_{\text{accumulation}}^{\text{MV}}$ in the lower panel. The times of cloud events are shown in red shadows.

To better understand the cloud effect of PNSDs, we compared the PNSDs at CVAO and MV during cloud events and non-

cloud events of different aerosol types. Fig. 8 shows the median PNSDs of different particle types during cloud events and





non-cloud events. During non-cloud events, PNSDs at CVAO ($PNSD^{CVAO}_{non-cloud}$) and MV ($PNSD^{MV}_{non-cloud}$) were similar for marine, mixture or dust type1 periods. During dust type2, there is only a very short period of non-cloud event with 15 PNSDs observed. Therefore, we did not include the comparison of $PNSD^{CVAO}_{non-cloud}$ and $PNSD^{MV}_{non-cloud}$ during dust type2 period in Fig. 8.

During non-cloud events, PNSDs at CVAO and MV were the same, as shown in Fig. 8. For periods with clouds, PNSDs in
5   the size range >80 nm at MV are lower than that at CVAO for all the particle types. For dust type 1 and dust type 2, depending on the clouds, i.e., the highest supersaturation the particles encounter, particles in Aitken mode were also activated to cloud droplets. For particles in the size range <40 nm, PNSDs are similar during cloud and non-cloud events. This is because the particle size is not large enough to activate to cloud droplet. Furthermore, it also indicates that PNSDs are similar at CVAO and MV during cloud events, at least in the size range <40 nm. For a more detailed comparison of PNSDs at CVAO and MV,
10  contour plots for PNSDs can be found in Fig. S9 in the supplement.

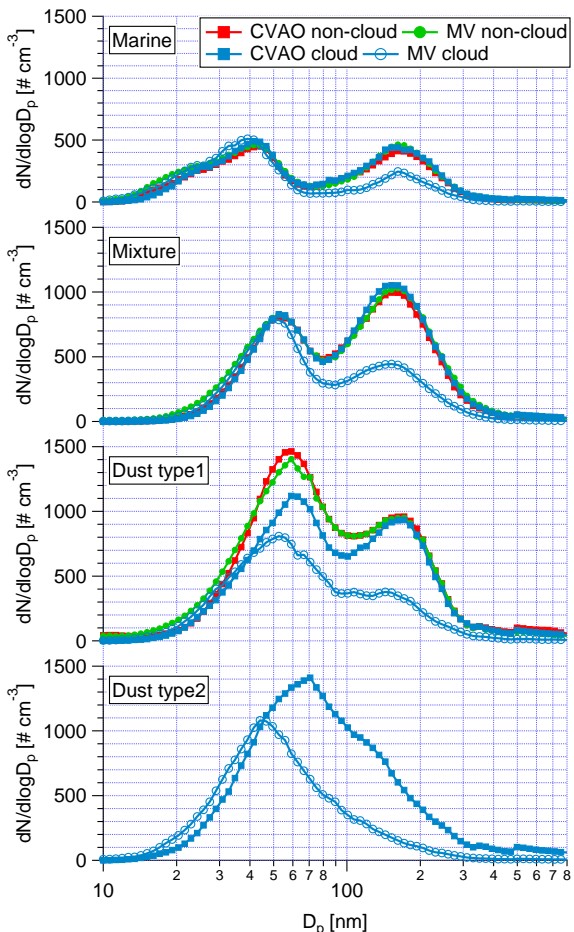

**Figure 8.** The median of PNSDs for 4 different particle types during cloud events and non-cloud events at CVAO and MV.



During the campaign, a decoupled marine boundary layer was observed with our balloon measurement in 3 cases, i.e., 10:30 to 11:00 16 September, 16:00 to 16:30 5 October and 17:20 to 17:50 12 October (shown as red dots in Fig. 1). Only for the first decoupling case (10:30 to 11:00 16 September), MV was cloud free, and nevertheless PNSDs were similar at CVAO and MV (Fig. S10). Therefore, the MBL may be generally well mixed, maybe still from times before the decoupling of the layers

formed. On the other hand, lifting of the air masses over the mountain might also partially explain this observation. However, due to the fact that there is only this one decoupled case, a thorough analysis of the influence of coupling and decoupling can not be done.

### 3.3.2  Comparison of $N_{\mathrm{CCN}}$

Fig. 9 shows the scatter plot of $N_{\mathrm{CCN}}$ at CVAO ($N_{\mathrm{CCN}}^{\mathrm{CVAO}}$) against that at MV ($N_{\mathrm{CCN}}^{\mathrm{MV}}$) during cloud events (green dots) and non-

cloud events (red rectangles) at different supersaturations. During cloud events, large particles that had been activated to cloud droplets, were removed by the aerosol inlet on MV. Therefore, $N_{\mathrm{CCN,cloud}}^{\mathrm{CVAO}}$ is larger than $N_{\mathrm{CCN,cloud}}^{\mathrm{MV}}$ at each supersaturation. During non-cloud events, all the data points are close the 1:1 line (for the slopes see Fig. 9) and $R^2$ between $N_{\mathrm{CCN,non-cloud}}^{\mathrm{CVAO}}$ and $N_{\mathrm{CCN,non-cloud}}^{\mathrm{MV}}$ are all above 0.90, indicating $N_{\mathrm{CCN}}$ is similar at CVAO and MV. Although there were slight differences in supersaturation at CVAO and MV due to the CCNc calibration, the similarity between $N_{\mathrm{CCN}}$ at the two stations conveys

the same message that what was discussed before, concerning the comparison of PNSDs at CVAO and MV, i.e., particles are generally well mixed in the MBL.

In short summary, in section 3.3, cloud events were observed at MV and can be identified based on the integrated concentrations between ground and cloud level. During the cloud events, larger particles (mainly accumulation and coarse mode) are activated to cloud droplets. Aitken mode particles starting with sizes of roughly 40 nm also can be activated to cloud droplets

if the cloud is strong enough. During non-cloud events, PNC, PNSD and $N_{\mathrm{CCN}}$ are similar at CVAO and MV. The aerosol particles measured at ground level (CVAO) can represent the aerosol particles at the cloud level (MV).

### 3.4  Particle hygroscopicity

In this section, we will focus on $N_{\mathrm{CCN}}$, $d_{\mathrm{crit}}$ and $\kappa$ measurements at CVAO. As outlined above, PNSDs and $N_{\mathrm{CCN}}$ measured at ground level are similar to that at cloud level. Therefore, measurements at the CVAO can be representative for that at MV.

Firstly, a thorough statistical analysis of $N_{\mathrm{CCN}}$, $d_{\mathrm{crit}}$ and $\kappa$ will be discussed. Secondly, the marine and dust aerosol particles' contribution to $N_{\mathrm{CCN}}$ and their $\kappa$ values will be compared.

### 3.4.1  Statistical analysis of $N_{\mathrm{CCN}}$, $d_{\mathrm{crit}}$ and $\kappa$

Fig. 10 shows the time series of $N_{\mathrm{total}}$ and $N_{\mathrm{CCN}}$ in the upper panel, $d_{\mathrm{crit}}$ in the middle panel and $\kappa$ in the lower panel, with different colors for different supersaturations. The error bars of $d_{\mathrm{crit}}$ show 1 standard deviation (std), and of $\kappa$ show 1

geometric standard deviation (geostd). Explanation of error bars can be found in section 2.4 as well as in the supplement. $N_{\mathrm{CCN}}$ shows large variability, e.g., $N_{\mathrm{CCN,0.30\%}}$ varied from 106 to 884 cm$^{-3}$, with a median of 509 cm$^{-3}$. We observed highest




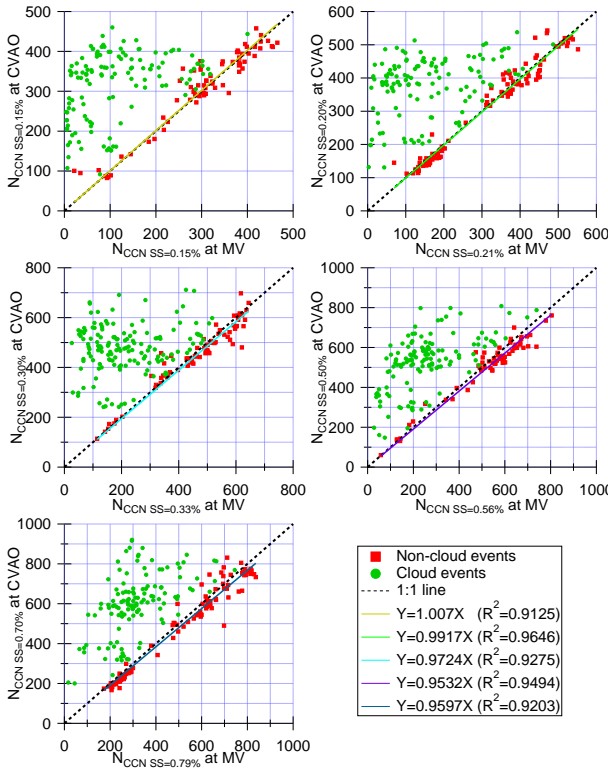

**Figure 9.** Scatter plots of $N_{CCN}$ at CVAO against those at MV at different supersaturations. Slope and $R^2$ for these fits are given in the panel.

$N_{CCN,0.30\%}$ of 503 cm$^{-3}$ (median) during dust type2 periods, and lowest $N_{CCN,0.30\%}$ of 109 cm$^{-3}$ (median) during marine periods. $N_{CCN,0.30\%}$ during different aerosol type periods are summarized in Tab. 2. Fig. 11(a) shows the box plot of $N_{CCN}$ at different supersaturations during the whole campaign. As can be seen, $N_{CCN}$ increases towards higher supersaturation, which is expected. The median of $N_{CCN}$ at different supersaturations also exhibited large variability, varying from 327 (median) at a supersaturation of 0.15% to 652 cm$^{-3}$ (median) at a supersaturation of 0.70%. Tab. 3 summarizes those numbers and shows additional details.

$d_{crit}$ at supersaturations of 0.15%, 0.20%, 0.50% and 0.70% were almost constant throughout the campaign, as shown in the middle panel in Fig. 10. The mean value of $d_{crit}$ and its std are summarized in Tab. 3. For the supersaturations of 0.70% and 0.50%, $d_{crit}$ is below 80 nm, i.e., inside the Aitken mode. However, for the lower supersaturations of 0.15% and 0.20%, $d_{crit}$ is located in the accumulation mode. Consequently, hygroscopicities derived at these supersaturations, can be assumed to be representative for the Aitken (at supersaturations of 0.70% and 0.50%) and the accumulation mode (at supersaturation of 0.10% and 0.20%), respectively. $d_{crit}$ at a supersaturation of 0.30% ($d_{crit,0.30\%}$) is not as constant as it is at other supersaturations, and it is larger during the marine type period than during other periods. With a median of 79.7 nm, it is close to the Hoppel minimum. Therefore, the hygroscopicity derived at a supersaturation of 0.30% can be assumed to be representative for the mixture of Aitken and accumulation particles.





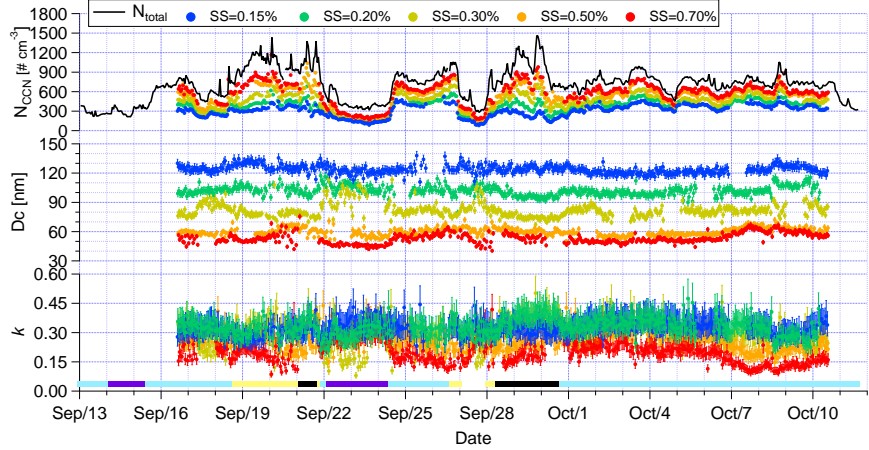

**Figure 10.** Time series of $N_{CCN}$ in the upper panel, $d_{crit}$ in the middle panel and $\kappa$ in the lower panel. All of those are measured at CVAO. Error bars of $d_{crit}$ and $\kappa$ show 1 standard deviation and 1 geometric standard deviation, respectively. The color bar in the lower panel indicates the times of different aerosol type periods. The meaning of different colors here is as same as in Fig. 4.

The particle hygroscopicity, expressed as $\kappa$, can be seen as a measure for average particle chemical composition. $\kappa$ values at different supersaturations show little variability over time (lower panel in Fig. 10), with geostd lower than 0.12, i.e., there is no clear trend in $\kappa$ over time during the campaign. A slightly increasing trend of $\kappa$ was observed with decreasing supersaturations, as shown in Fig. 11(c). At supersaturations of 0.70% and 0.50%, i.e., for Aitken mode particles, $\kappa$ values are 0.18 and 0.25

5  (geomean), respectively. At the lowest supersaturation of 0.15% and 0.20%, i.e., for accumulation mode particles, $\kappa$ values are 0.32 and 0.34 (geomean). Tab. 3 summarizes those numbers and shows additional details.

Fig. 11(d) shows $\kappa$ as a function of $d_{crit}$ and error bars of $\kappa$ and $d_{crit}$ show geostd and std, respectively. A slightly increasing trend of $\kappa$ over increasing $d_{crit}$ is observed. It suggests that the soluble, likely inorganic material added during cloud processing increases $\kappa$ of the originally very organic rich particles, which has also been observed in previous studies (Kalivitis et al., 2015;

10  Kristensen et al., 2016). Overall, $\kappa$ averaged 0.28. Pringle et al. (2010) used an atmospheric chemistry model to derive global distributions of effective particle hygroscopicity $\kappa$. For CVAO this model resolved an annual cycle of monthly mean $\kappa$ value ranged from 0.25 in February to 0.60 in April. For September and October, the period of this study, the value of 0.35 and 0.30 was reported, respectively, which is consistent with what we obtained during this campaign.

The Hoppel minimum diameter range of 70 to 100 nm for different aerosol types (mentioned in section 3.2.2), together

15  with the average $\kappa$ of 0.28, can be used to obtain a rough estimate of maximum supersaturations present in trade wind clouds along the path of the sampled air masses. Resulting values are roughly 0.22% to 0.37% for dust type2 and marine air masses, respectively. This is close to an earlier estimate given in Clarke et al. (1996) of 0.35% and can be interpreted as typical value for trade wind cumuli.





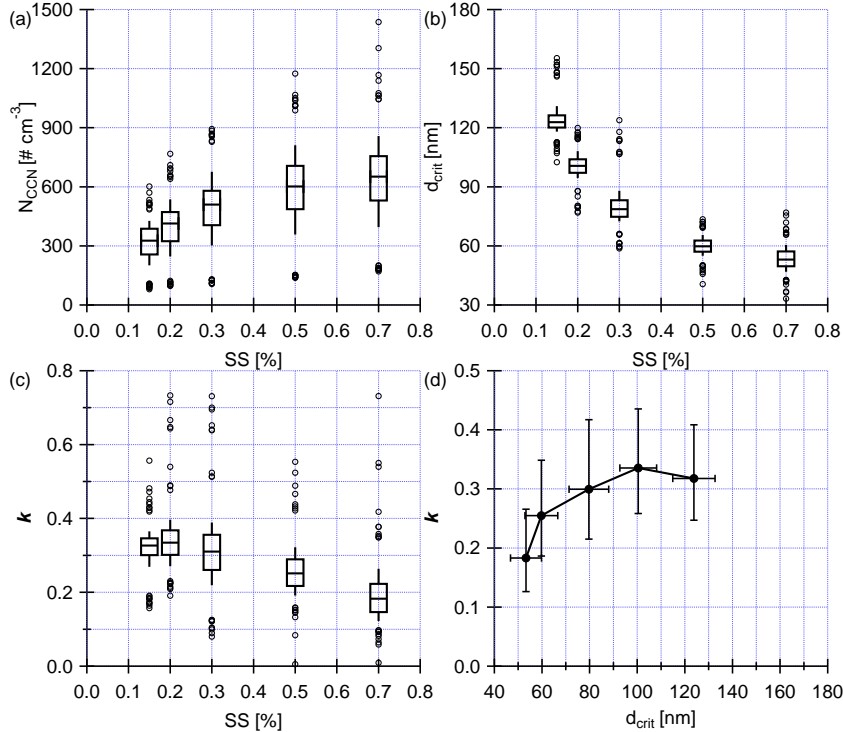

**Figure 11.** Boxplot of $N_{CCN}$ (a), $d_{crit}$ (b) and $\kappa$ (c) at different supersaturations. Whiskers show the 10% to 90% percentiles. Circles show the outliers (1%). (d) $\kappa$ as a function of $d_{crit}$. Error bars of $d_{crit}$ and $\kappa$ show 1 standard deviation and 1 geometric standard deviation, respectively.

### 3.4.2 Dust and marine comparison

In this section, we will focus on examining the difference between the cleanest periods (marine type) and heaviest observed dust pollution periods (dust type2). Therefore, we compared $N_{CCN}$ and $\kappa$ during marine type and dust type2 periods. Fig. 12 shows the box plot of $N_{CCN}$ as a function of supersaturation in the upper panel. As outlined above, during dust periods, the

5   aerosol shows a great enhancement in the Aitken, accumulation and coarse mode particles, therefore, overall $N_{CCN}$ increases at different supersaturations. It is clear that $N_{CCN}$ during dust type2 periods is much higher than that during marine periods. For example, $N_{CCN,0.30\%}$ were 503 and 190 (median) cm$^{-3}$ during dust type2 and marine periods, respectively. During marine periods, $N_{coarse}$, i.e., SSA particles, accounted for roughly 3.7% of $N_{CCN,0.30\%}$, for the range of wind speeds from 4 to 10 m s$^{-1}$ that were present during this study. This is relatively low compare to e.g., Wex et al. (2016) who found that the SSA particles

10   accounted for up to 15% of $N_{CCN,0.30\%}$ for wind speeds up to 14 m s$^{-1}$ for the marine aerosol on Barbados, and Modini et al. (2015) who found that SSA particles accounted for up to 16% to 28% (wind speeds up to 16 m s$^{-1}$) and 5% to 10% (wind speed from 4 to 10 m s$^{-1}$) of $N_{CCN,0.30\%}$. Therefore the lower fractions of SSA particles in our study are likely connected to the low wind speeds we encountered.



**Table 3.** Median and mean values of $N_{CCN}$, $d_{crit}$, $\kappa$, one standard deviation of $d_{crit}$ and one geometric standard deviation of $\kappa$ at different supersaturations.

| | Supersaturation (%) | $N_{CCN}$ (cm$^{-3}$) | $d_{crit}$ (nm) | $\kappa$ |
|---|---|---|---|---|
| | | median, mean, ±std | mean±std | geomean±geostd [*] |
| Whole campaign | 0.15 | 327, 320±88 | 123.8±8.9 | 0.32±0.09, 0.07 |
| | 0.20 | 414, 400±112 | 100.4±7.7 | 0.34±0.10, 0.08 |
| | 0.30 | 509, 495±143 | 79.7±8.4 | 0.30±0.12, 0.08 |
| | 0.50 | 602, 593±176 | 59.8±6.9 | 0.25±0.09, 0.07 |
| | 0.70 | 652, 638±186 | 53.3±6.5 | 0.18±0.08, 0.06 |
| Marine | 0.15 | 146, 155±37 | 121.2±5.0 | 0.34±0.08, 0.06 |
| | 0.20 | 166, 177±43 | 103.5±7.8 | 0.31±0.10, 0.08 |
| | 0.30 | 190, 202±49 | 87.8±15.7 | 0.23±0.17, 0.10 |
| | 0.50 | 191, 215±70 | 56.9±5.7 | 0.30±0.12, 0.09 |
| | 0.70 | 235, 260±72 | 46.1±2.2 | 0.28±0.06, 0.05 |
| Dust type2 | 0.15 | 259, 242±56 | 124.6±4.7 | 0.32±0.07, 0.06 |
| | 0.20 | 370, 357±77 | 96.9±4.2 | 0.37±0.08, 0.07 |
| | 0.30 | 503, 501±105 | 74.9±3.6 | 0.36±0.08, 0.06 |
| | 0.50 | 654, 636±125 | 60.4±3.2 | 0.25±0.06, 0.05 |
| | 0.70 | 798, 764±111 | 52.6±5.5 | 0.19±0.08, 0.06 |

[*] one geometric standard deviation

$\kappa$ as a function of $d_{crit}$ is shown in the lower panel in Fig. 12(b). The error bars of $d_{crit}$ and $\kappa$ show std and geostd, respectively. During dust type2, slightly increasing $\kappa$ with increasing $d_{crit}$ was observed, similar to the overall trend we described above. $\kappa$ featured lower values from 0.13 to 0.31 for Aitken mode particles, while higher values from 0.26 to 0.45 were found for accumulation mode particles. Until now, the only field measurement of particle hygroscopicity during a dust plume at Cape

5 Verde was carried out by Schladitz et al. (2011a), who found that hygroscopic particles featured a $\kappa$ value (based on hygroscopic growth factor of particles) from 0.35 to 0.65. Our CCN derived $\kappa$ values in this study for the aerosol influenced by dust are therefore comparable to the values reported by Schladitz et al. (2011a).

No clear trend of $\kappa$ with $d_{crit}$ was observed during marine type periods (as shown in Fig. 12), during which $\kappa$ averaged 0.30. Larger error bars of $\kappa$ and $d_{crit}$ at the supersaturation of 0.30% were observed, as in this case the $d_{crit}$ is close to the Hoppel

10 minimum where a small change in $N_{CCN}$ causes a comparably large change in $d_{crit}$ (explained in the supplement). Kristensen et al. (2016) found that for the marine aerosol on Barbados in June and July 2013 values for $\kappa$ of 0.2 to 0.5 were derived, which is consistent with this study. When considering the scatter observed in $\kappa$ (see the error bars in Fig. 12), $\kappa$ during the dust

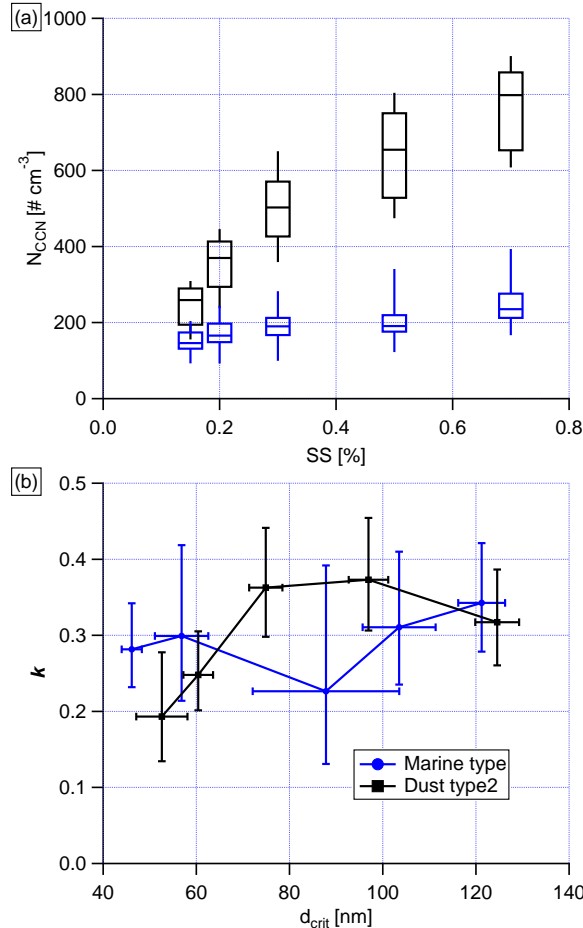

**Figure 12.** (a) Boxplot of $N_{CCN}$ as a function of $\kappa$ for marine type (blue) and dust type2 (black). Whiskers show the 10% to 90% percentiles. Circles show the outliers (1%). (b) $\kappa$ as a function of $d_{crit}$ for marine type (blue) and dust type2 (black). Error bars of $d_{crit}$ and $\kappa$ show 1 standard deviation and 1 geometric standard deviation, respectively.

type2 period still agreed with that of the marine period within uncertainty. Therefore, no distinguishable difference of $\kappa$ during marine and dust periods in the size range from 40 to 140 nm were found during this campaign.

In short summary of section 3.4, overall, there is a slight increase of $\kappa$ with particle size, indicating the addition of soluble, likely inorganic material during cloud processing. $\kappa$ values in this study are comparable to previous model work and field measurement. $N_{CCN}$ during the heaviest observed dust periods is much higher than that during marine periods, while $\kappa$ values for these two periods show no significant difference.



## 4   Conclusions

The MarParCloud campaign took place in September and October 2017 on the Cape Verde islands to investigate the aerosols prevailing in the Atlantic Ocean. As the first in a series of publications to come from the MarParCloud campaign, this study provides a thorough characterization of the abundance, properties, and sources of aerosol particles in general and CCN in particular close to both sea and cloud level heights with measurements done at the Cape Verde Atmospheric Observatory (CVAO) and on the top of Monte Verde (MV), respectively.

$N_{\text{total}}$ varied from ∼200 to ∼1500 cm$^{-3}$, with a median of ∼700 cm$^{-3}$ at CVAO. A trimodal parameterization method was deployed to characterize PNC. Based on number concentrations in different aerosol modes, four well separable types of PNSDs were found, i.e., the marine type, mixture type, dust type1 and dust type2. These different aerosol types originate from different regions. The marine type aerosol mainly originates from the Atlantic Ocean, while the dust type aerosol mainly comes from the Saharan region. During marine periods, the coarse mode can be attributed to sea spray aerosol, and the corresponding particle number concentration accounted about 3.7% of $N_{\text{CCN,0.30\%}}$ and about 1.1% to 4.4% of $N_{\text{total}}$. Because of lower wind speeds that were present at CVAO, this value is lower than previous field measurement (Modini et al., 2015; Wex et al., 2016).

A thorough comparison of PNC, PNSDs and $N_{\text{CCN}}$ at CVAO and MV clearly showed these parameters to be similar at both stations in the absence of clouds. Cloud events were observed at MV during roughly 58% of the time. During the cloud events, larger particles (mainly accumulation and coarse mode) are activated to cloud droplets and our data suggests that the maximum supersaturation in the clouds is higher the cleaner the air mass gets, leading to a lower Hoppel minimum. Altogether, it was observed that the boundary layer is generally well mixed, therefore CVAO can be used to describe the aerosol particles at cloud level.

Overall, $\kappa$ averaged 0.28, suggesting the presence of organic material in particles. This is consistent with previous model work (Pringle et al., 2010) and field measurement of hygroscopic growth (Schladitz et al., 2011a) done for the location of Cape Verde. There is a slight increase of $\kappa$ with particle size, indicating the addition of soluble, likely inorganic material during cloud processing. When looking at the two most different aerosol types, the marine type and dust type2, $\kappa$ values for these periods show no significant difference. On the other hand, dust plumes enhanced particle concentrations in the Aitken, accumulation and coarse mode and therefore, overall increased $N_{\text{CCN}}$. $N_{\text{CCN,0.30\%}}$ during the strongest observed dust periods is about 2.5 times higher than that during marine periods.

*Data availability.*   The data are available through the World Data Center PANGAEA (https://www.pangaea.de/) in the near future. A link to the data can be found under this paper's assets tab on ACP's journal website.

*Author contributions.*   X. Gong wrote the manuscript with contributions from J. Voigtländer, H. Wex and M. van Pinxteren. K. Weinhold and X. Gong performed MPSS and APS measurements and X. Gong performed data evaluation. K. Weinhold calibrated MPSS and APS before, during and after the campaign. F. Stratmann, H. Wex and X. Gong performed the CCN measurements and X. Gong performed data evaluation.



S. Henning calibrated CCN before and after the campaign. Balloon measurements and data evaluation were performed by J. Voigtländer and X. Gong. X. Gong, H. Wex and F. Stratmann discussed the results and further analysis after the campaign. All co-authors proofread and commented the manuscript.

*Competing interests.*  The authors declare that they have no conflict of interests.

5  *Acknowledgements.*  The works were carried out in the framework of the MarParCloud project. The authors acknowledge the Leibniz Association SAW funding for the project "Marine biological production, organic aerosol particles and marine clouds: a Process Chain (MarParCloud)", SAW-2016-TROPOS-2. We are grateful to Dr. Lucy Carpenter and Dr. Katie Read, from the University of York and the Atmospheric Measurement and Observation Facility at the National Centre for Atmospheric Science (NCAS-AMOF), for kindly providing the meteorology data at CVAO.



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
