# Peer review of "Characterization of aerosol particles at Cape Verde close to sea and cloud level heights - Part 1: particle number size distribution, cloud condensation nuclei and their origins"

_Atmospheric Chemistry and Physics, 2019_

## Referee Comment (RC1) · Anonymous Referee #1 · 12 Sep 2019

The paper presents a very straightforward description of aerosol measurements taken on Cape Verde during September and October of 2017. Particle and CCN number concentrations and particle size distributions are reported and compared for marine and dust aerosol types. The data presented are largely consistent with previously reported results for the North Atlantic. The few minor concerns I have are listed below.

Page 3, lines 4 – 6: Please define what is meant by "marine" aerosol. Given this statement on production mechanisms it appears to include all ocean-derived aerosol, not just sea spray.

[Figure]

Page 11, line 8: Cloud processing should also be included here in the growth of accumulation mode particles (e.g., Hoppel et al., JGR, vol. 99, p. 14443, 1994).

Page 13, lines 2 – 4: The criteria of "marine type" as having an Aitken mode larger than the accumulation mode needs more justification. What about cases when cloud cycling has occurred allowing for more accumulation of mass? Were the trajectories also used to categorize the aerosol types?

Page 15, lines 2 – 5: Is it possible that the high Aitken number concentrations observed during the dust episodes could also be a result of input from the upper troposphere in the boundary layer? Figure 6 does not provide information about the vertical path of the calculated back trajectories.

Figure 10 caption: It would be helpful to include the color bar information here so the reader does not have to refer back to Figure 4.

Page 20, lines 11 – 13: Please provide a brief description of causes of the seasonality in kappa.

Page 21, lines 11 – 12: Fractions of SSA also are a function of the amount of non-SSA present. As stated earlier in the paper, the number concentration of SSA was not fully explained by local wind speed.

---

## Referee Comment (RC2) · Anonymous Referee #2 · 1 Nov 2019

The manuscript discusses aerosol size distributions and CCN measurements made at multiple supersaturations observed at Cape Verde as part of the MarParCloud project. Measurements were conducted at two stations – a low-altitude, coastal station (CVAO) and a higher-altitude, mountainous station (MV), with the latter station being located close to cloud level. The size distributions are fit to 2-3 lognormal modes (Aitken, accumulation, coarse), and very rough inferences of aerosol type are drawn from relative fractional contributions of these modes as well as HYSPLIT air mass backtrajectories. CCN measurements show size-dependent hygroscopicity with lower kappas observed

at higher supersaturations (with the implication that these measurements are representative of smaller aerosol sizes). This leads to the interesting conclusion that cloud processing both transitions aerosol from the Aitken to accumulation mode, but also slightly increases the kappa. Overall, the paper is well written and relevant to ACP. I recommend publication after the following comments are satisfactorily addressed:

1) The paper as it is currently written stands on its own, and from the brief description given of the second paper, it also sounds like it too will adequately stand on its own. Consequently, I recommend that the title be revised so that this is not be a two-part paper. Alternatively, the authors should provide a copy of the companion manuscript and explain why the two are inextricably linked.

2) The language and concept of deploying a "trimodal parameterization method" as described in the abstract and elsewhere (e.g., Pg. 9, Lines 2-3; Pg. 24, Lines 7-8) implies that something novel has been developed, which is not the case. The size distribution measurements are fit to multiple lognormal functions to derive summary statistical parameters, using fit functions that are textbook and commonplace. Please revise this language to indicate that the "parameterization method" is actually "fitting the data to multiple lognormal functions".

3) How were the size modes and backtrajectory information synthesized to arrive at the four aerosol type classifications in the present paper? Would it make more sense to conform to the 5-type classification scheme of Fomba et al. (2014)?

4) The sentence on Pg. 2, Line 14 is awkward and unnecessary. I suggest it be removed.

5) Pg. 2, Ln. 22-23: Karydis et al. (2011) did not find that dust contributes up to 40% to CCN on a global basis. This was found for the N. African and Asian desert regions.

6) Pg. 3, Ln. 9: remove "besides"

7) Pg. 3, Ln. 12: Quinn et al. (2017) did not find that "marine aerosol" contributes

less than 30% to CCN. They use the term "sea spray aerosol", and suggest that SSA contributed less than 30% to CCN. Organics and secondary sulfate of marine origin can dominate CCN in remote regions.

8) Pg. 3, Ln. 16: Something is amiss with the total mass reported of 47.2 +/- 55.5, as it implies substantial negative mass ($\sim$ -8.3 ug/mˆ3). I suspect that the observations here lack normality and the use of an arithmetic mean and standard deviation is inappropriate.

9) Pg. 3, Ln. 31 (and multiple instances elsewhere): The use of the phrase "to the best of our knowledge,...", is sloppy writing and gives the reader the impressions that the authors have not done their due diligence in conducting a literature survey. If the statement is true (which I think it is), then it should stand on its own without the need for such a caveat.

10) Pg. 3, Ln. 31: "filed" = "field"

11) Pg. 4, Ln. 18: "see" = "sea"

12) Pg. 4, Ln. 20-21: Is it really the first time these measurements have been conducted in Cape Verde? Why is the "to the best of our knowledge" caveat here?

13) Pg. 4, Ln. 25: Please update reference or remove it if the paper is still in preparation.

14) Pg. 4, Ln. 30-31: Are the winds always from the northeast?

15) Pg. 4, Ln. 31-33: Please add citations to support these sentences related to annual rainfall and precipitation even frequency.

16) Pg. 7, Ln. 3-4: How was the APS data used to correct the MPSS data for multiple charges as the APS is measurement aerodynamic diameter? What assumptions were invoked?

17) Pg. 7, Ln. 5: "base" = "basis"

18) Pg. 7, Ln 11: Please add a sentence to the end of this paragraph summarizing how approximately how large the particle loss corrections ended up being (e.g., on the order of 10%, something smaller, or something larger?).

19) Pg. 10, Ln. 2: What is meant by "behavior of aerosols" here? Is this discussed in this manuscript?

Table 1: Please reformat the table so the Measurement Site and Location fields are on the same line as the other information.

Table 3: I don't understand what is being presented in the kappa column. Is one of the numbers the + and the other the -? If so, which is which. Would it be better to report the geomean */ geostd?

Figure 12: It would be really interesting to use the median size distributions from Fig. 5 to compute and overlay lines of constant kappa for each case to evaluate how the box-whiskers fall across the range of hygroscopicities.

———————————————

---

## Author Comment (AC1) · 14 Nov 2019

Dear Reviewer,

We thank you for doing this review and for your suggestions that helped to improve our manuscript. Below, please find your original comments in blue and our responses in black. When referencing page and line numbers, we are always referring to the original versions of manuscript and SI.

The paper presents a very straightforward description of aerosol measurements taken on Cape Verde during September and October of 2017. Particle and CCN number concentrations and particle size distributions are reported and compared for marine and dust aerosol types. The data presented are largely consistent with previously reported results for the North Atlantic. The few minor concerns I have are listed below.

Page 3, lines 4 – 6: Please define what is meant by "marine" aerosol. Given this statement on production mechanisms it appears to include all ocean-derived aerosol, not just sea spray.

Thanks for your comment. We added the following in page 3, line 4:

"Together with newly formed particles originating from gaseous precursors which can also be emitted from the ocean, this sea spray aerosol (SSA) contributes to marine aerosols."

Page 11, line 8: Cloud processing should also be included here in the growth of accumulation mode particles (e.g., Hoppel et al., JGR, vol. 99, p. 14443, 1994).

Thanks for your comment. We changed page 11, lines 6-9 and added your comments:

"Generally, Aitken mode particles are produced by homogeneous and heterogeneous nucleation processes, formed during natural gas-to particle condensation. Aitken mode particles are transferred to the accumulation mode through cloud processing (Hoppel et al., 1994) and accumulation mode particles are furthermore formed by coagulation of smaller particles or condensation of vapors onto existing particles, during which they grow into that size range (Seinfeld and Pandis, 2016)."

 The criteria of "marine type" as having an Aitken mode larger than the accumulation mode needs more justification. What about cases when cloud cycling has occurred allowing for more accumulation of mass? Were the trajectories also used to categorize the aerosol types?

We tried different criteria for particle classification and checked our classification with backward trajectories.

When times with $N_{Aitken} < N_{accumulation}$ were included in the marine type aerosol, also trajectories that had passed closer to or over Africa were added to this mode, i.e., such a clear distinction for the trajectories as the one shown in the current manuscript could only be found when the criteria $N_{Aitken} > N_{accumulation}$ was included in the analysis. Therefore, it could be said that trajectories were also used for the categorization.

We added the following in page 13, line 4 to clarify:

"For the separation of this marine type, additionally also trajectories were examined."

 Is it possible that the high Aitken number concentrations observed during the dust episodes could also be a result of input from the upper troposphere in the boundary layer? Figure 6 does not provide information about the vertical path of the calculated back trajectories.

Thanks for your comment. Indeed, during the dust type2 period, we observed that the backward trajectories often traveled from ~2500 m to the marine boundary. Therefore, the Aitken number concentration could also be a result of input from the upper troposphere in the marine boundary layer. We added the following to clarify:

"In our data, we found that backward trajectories often travelled from the upper troposphere down to the marine boundary during dust periods, which means that Aitken mode particles could have been transported from the upper troposphere. Therefore, there are different factors contributing to the observed high $N_{Aitken}$ and $N_{accumulation}$ during dust plumes, such as direct transport of particles from the desert and Sahel region, and additional new particle formation and growth in the vicinity or in the upper troposphere."

Figure 10 caption: It would be helpful to include the color bar information here so the reader does not have to refer back to Figure 4.

Done.

Page 20, lines 11 – 13: Please provide a brief description of causes of the seasonality in kappa.

We added the following in page 20, line 12:

 "This annual circle of $\kappa$ likely originated in a change of chemical composition of the aerosol throughout the year, related to different precursors and a higher organic content during times with higher algal activity."

Page 21, lines 11 – 12: Fractions of SSA also are a function of the amount of non-SSA present. As stated earlier in the paper, the number concentration of SSA was not fully explained by local wind speed.

Thanks for your comment. We did indeed not acknowledge a possible higher fraction of particles in the Aitken and accumulation modes, and now mention that this impedes a direct comparison of the here discussed fractions. We revised page 21, lines 12-13 such that this should be clear now:

[revised manuscript text omitted]

The dry density of Saharan dust particles was determined in a range of $\rho$ = 2450 - 2700 kg m$^{-3}$ over the Cape Verde Islands (Haywood et al., 2001). The dry particle density of sodium chloride is known to be $\rho$ = 2160 kg m$^{-3}$. The overall effective density of the dust and sea-salt fraction is approximately 2, as recommened in Schladitz et al. (2011).

The dry dynamic shape factor $\chi$ of mineral dust is $\chi$ = 1.25 (Kaaden et al., 2009) for 1 $\mu$m particles, whereas the dynamic shape factor for sodium chloride is $\chi$ = 1.08 (Kelly and McMurry, 1992; Gysel and Stratmann, 2013). We used the average shape factor of 1.17 in this study.

Based on these, a conversion from aerodynamic to geometric diameters were done for the APS data, and particle number concentrations from the APS were used to correct the multiply charged particle concentrations in the upper size range where the MPSS measured.

[revised manuscript text omitted]

10   Atmos. Chem. Phys., 18, 4477–4496, https://doi.org/10.5194/acp-18-4477-2018, https://www.atmos-chem-phys.net/18/4477/2018/, 2018.

Kaaden, N., Massling, A., Schladitz, A., Müller, T., Kandler, K., Schütz, L., Weinzierl, B., Petzold, A., Tesche, M., Leinert, S., Deutscher, C., Ebert, M., Weinbruch, S., and Wiedensohler, A.: State of mixing, shape factor, number size distribution, and hygroscopic growth of the Saharan anthropogenic and mineral dust aerosol at Tinfou, Morocco, Tellus B, 61, 51–63, https://doi.org/doi:10.1111/j.1600-0889.2008.00388.x, https://onlinelibrary.wiley.com/doi/abs/10.1111/j.1600-0889.2008.00388.x, 2009.

15   Kelly, W. P. and McMurry, P. H.: Measurement of Particle Density by Inertial Classification of Differential Mobility Analyzer–Generated Monodisperse Aerosols, Aerosol Science and Technology, 17, 199–212, https://doi.org/10.1080/02786829208959571, https://doi.org/10.1080/02786829208959571, 1992.

Kristensen, T. B., Müller, T., Kandler, K., Benker, N., Hartmann, M., Prospero, J. M., Wiedensohler, A., and Stratmann, F.: Properties of cloud condensation nuclei (CCN) in the trade wind marine boundary layer of the western North Atlantic, Atmos. Chem. Phys., 16, 2675–2688,

20   https://doi.org/10.5194/acp-16-2675-2016, http://www.atmos-chem-phys.net/16/2675/2016/, 2016.

Schladitz, A., Müller, T., Nowak, A., Kandler, K., Lieke, K., Massling, A., and Wiedensohler, A.: In situ aerosol characterization at Cape Verde, Part1: Particle number size distributions, hygroscopic growth and state of mixing of mrine and Saharan dust aerosol, Tellus B, 63, 531–548, https://doi.org/10.1111/j.1600-0889.2011.00569.x, http://dx.doi.org/10.1111/j.1600-0889.2011.00569.x, 2011.

von der Weiden, S. L., Drewnick, F., and Borrmann, S.: Particle Loss Calculator - a new software tool for the assessment of the performance

25   of aerosol inlet systems, Atmos. Meas. Tech., 2, 479–494, https://doi.org/10.5194/amt-2-479-2009, http://www.atmos-meas-tech.net/2/479/2009/, 2009.

---

## Author Comment (AC2) · 14 Nov 2019

Dear Reviewer,

We thank you for doing this review and for your suggestions that helped to improve our manuscript. Below, please find your original comments in blue and our responses in black. When referencing page and line numbers, we are always referring to the original versions of manuscript and SI.

The manuscript discusses aerosol size distributions and CCN measurements made at multiple supersaturations observed at Cape Verde as part of the MarParCloud project. Measurements were conducted at two stations – a low-altitude, coastal station (CVAO) and a higher-altitude, mountainous station (MV), with the latter station being located close to cloud level. The size distributions are fit to 2-3 lognormal modes (Aitken, accumulation, coarse), and very rough inferences of aerosol type are drawn from relative fractional contributions of these modes as well as HYSPLIT air mass backtrajectories. CCN measurements show size-dependent hygroscopicity with lower kappas observed at higher supersaturations (with the implication that these measurements are representative of smaller aerosol sizes). This leads to the interesting conclusion that cloud processing both transitions aerosol from the Aitken to accumulation mode, but also slightly increases the kappa. Overall, the paper is well written and relevant to ACP. I recommend publication after the following comments are satisfactorily addressed:

1) The paper as it is currently written stands on its own, and from the brief description given of the second paper, it also sounds like it too will adequately stand on its own. Consequently, I recommend that the title be revised so that this is not be a two-part paper. Alternatively, the authors should provide a copy of the companion manuscript and explain why the two are inextricably linked.

The two manuscripts are inextricable connected to each other. Part 2 is under discussion on ACPD. Please refer to this link for more details. https://www.atmos-chem-phys-discuss.net/acp-2019-729/

The two companion manuscripts both belong to the MarParCloud project and discuss the aerosol present at Cape Verde from 13 September to 13 October, where both manuscripts examine data from the CVAO at sea level and from the Monte Verde. Part 1 focuses on aerosol particle number size distribution (PNSD) and CCN number concentration and Part 2 focuses on ice nucleating particles (INPs). Part 2 does somewhat rely on Part 1, as it is important that we observed, in Part

1, that the marine boundary layer is well mixed, which cannot be deduced for INP (due to the cloud events, which only enable to get a few INP filter samples from Monte Verde). We would therefore prefer to keep the titles as they are now.

2) The language and concept of deploying a "trimodal parameterization method" as described in the abstract and elsewhere (e.g., Pg. 9, Lines 2-3; Pg. 24, Lines 7-8) implies that something novel has been developed, which is not the case. The size distribution measurements are fit to multiple lognormal functions to derive summary statistical parameters, using fit functions that are textbook and commonplace. Please revise this language to indicate that the "parameterization method" is actually "fitting the data to multiple lognormal functions".

The "trimodal parameterization method" is a well-known method to fit the particle number size distribution. We cited the textbook (Seinfeld and Pandis, 2016) in the manuscript.

For clarification, we changed page 9, lines 2-3 to:

"A well-known trimodal log-normal parameterization method is adopted to characterize the temporal variation of PNC in three modes."

We changed page 10, lines 10-11 to:

"To better define the modes of our data, we fitted the PNSDs to three log-normal functions."

3) How were the size modes and backtrajectory information synthesized to arrive at the four aerosol type classifications in the present paper? Would it make more sense to conform to the 5-type classification scheme of Fomba et al. (2014)?

The particle classification was based on the particle number concentration in different modes. The classification criteria was summarized in Tab. 2. The backward trajectories (1-hour resolution) were used for characterizing the different particle sources. In other word, the backward trajectories were used to double check if our particle classification was reasonable or not. Theoretically, if we classified the particle sources based on backward trajectories and checked the particle size distribution for different sources, the results should be similar to this paper. We do think the PNSD

measurements are more precise than the backward trajectories; therefore, we used the PNSD to do particle classification rather than used the backward trajectories.

Fomba et al. (2014) classified the particle sources from 2007 to 2011 at CVAO according to the backward trajectories (24-hour resolution) and then discussed the chemical composition difference for different particle sources. This manuscript focused on particle number concentration and Fomba et al. (2014) focused on particle mass concentration. Fomba et al. (2014) carried out 5-year measurement, which means more aerosol types were found. For example, Fomba et al. (2014) found that the $PM_{10}$ could reach ~200 µg/m$^{-3}$ during heavy dust periods, while in this study, the $PM_{10}$ was at most about ~40 µg/m$^{-3}$ during dust type 2 period. A comparison of particle classification between mass concentration and number concentration is discussed in the MarParCloud project overview paper (submitted to ACPD).

4) The sentence on Pg. 2, Line 14 is awkward and unnecessary. I suggest it be removed.

Done.

5) Pg. 2, Ln. 22-23: Karydis et al. (2011) did not find that dust contributes up to 40% to CCN on a global basis. This was found for the N. African and Asian desert regions.

Thanks for your comment. It was changed to:

"Karydis et al. (2011) found that the predicted annual average contribution of insoluble mineral dust to CCN number concentration in cloud forming areas is up to 40% over North Africa and Asia (Arabian Peninsula and Gobi Desert)."

6) Pg. 3, Ln. 9: remove "besides"

Done.

7) Pg. 3, Ln. 12: Quinn et al. (2017) did not find that "marine aerosol" contributes less than 30% to CCN. They use the term "sea spray aerosol", and suggest that SSA contributed less than 30% to CCN. Organics and secondary sulfate of marine origin can dominate CCN in remote regions.

The first reviewer suggested defining "marine aerosol" in the manuscript. We add the following in page 3, line 4:

"Together with newly formed particles originating from gaseous precursors which can also be emitted from the ocean, this sea spray aerosol (SSA) contributes to marine aerosols.

Therefore, in page 3, line 12, we changed it to:

"On a global basis, SSA makes a contribution of less than 30% to the CCN population (Quinn et al., 2017)."

8) Pg. 3, Ln. 16: Something is amiss with the total mass reported of 47.2 +/- 55.5, as it implies substantial negative mass (~ -8.3 ug/m^3). I suspect that the observations here lack normality and the use of an arithmetic mean and standard deviation is inappropriate.

Yes, you are right. The standard deviation is larger than the mean, implying that the data is not normally distributed for a strictly positive data set. It would be better to use median and interquartile range to represent of the data. However, only mean, median and standard deviation were reported in Fomba et al. (2014).

9) Pg. 3, Ln. 31 (and multiple instances elsewhere): The use of the phrase "to the best of our knowledge,...", is sloppy writing and gives the reader the impressions that the authors have not done their due diligence in conducting a literature survey. If the statement is true (which I think it is), then it should stand on its own without the need for such a caveat.

We removed "To the best of our knowledge" in the manuscript.

10) Pg. 3, Ln. 31: "filed" = "field"

Done.

11) Pg. 4, Ln. 18: "see" = "sea"

Done.

12) Pg. 4, Ln. 20-21: Is it really the first time these measurements have been conducted in Cape Verde? Why is the "to the best of our knowledge" caveat here?

We removed "To the best of our knowledge" in the manuscript.

13) Pg. 4, Ln. 25: Please update reference or remove it if the paper is still in preparation.

The overview paper is submitted to ACPD. This information will be updated in the new version.

14) Pg. 4, Ln. 30-31: Are the winds always from the northeast?

Yes, they are. The Cape Verde islands receive north-easterly prevailing trade winds blowing directly off the ocean. This is described in Carpenter et al. (2010).

15) Pg. 4, Ln. 31-33: Please add citations to support these sentences related to annual rainfall and precipitation even frequency.

Done. We cited Fomba et al. (2013) and Carpenter et al. (2010).

16) Pg. 7, Ln. 3-4: How was the APS data used to correct the MPSS data for multiple charges as the APS is measurement aerodynamic diameter? What assumptions were invoked?

The dry density of Saharan dust particles was determined in a range of $\rho = 2450 - 2700$ kg m$^{-3}$ over the Cape Verde Islands (Haywood et al., 2001). The dry particle density of sodium chloride is known to be $\rho = 2160$ kg m$^{-3}$. The overall effective density of the dust and sea-salt fraction is approximately 2, as recommended in Schladitz et al. (2011).

The dry dynamic shape factor $\chi$ of mineral dust is $\chi = 1.25$ (Kaaden et al., 2009) for 1 μm particles, whereas the dynamic shape factor for sodium chloride is $\chi = 1.08$ (Kelly and McMurry, 1992;Gysel et al., 2002). We used the average shape factor of 1.17 in this study.

Based on these, a conversion from aerodynamic to geometric diameters were done for the APS data, and particle number concentrations from the APS were used to correct the multiply charged particle concentrations in the upper size range where the MPSS measured.

Above information will be mentioned in the supplement.

17) Pg. 7, Ln. 5: "base" = "basis"

Done.

18) Pg. 7, Ln 11: Please add a sentence to the end of this paragraph summarizing how approximately how large the particle loss corrections ended up being (e.g., on the order of 10%, something smaller, or something larger?).

We added: "Overall, less than 3% of the particles were lost when passing the inlet."

19) Pg. 10, Ln. 2: What is meant by "behavior of aerosols" here? Is this discussed in this manuscript?

This might have been a misleading formulation and we removed it such that the sentence now is:

"Particle size is one of the most important parameters to characterize the atmospheric aerosol."

Table 1: Please reformat the table so the Measurement Site and Location fields are on the same line as the other information.

Table 1 was changed.

Table 3: I don't understand what is being presented in the kappa column. Is one of the numbers the + and the other the -? If so, which is which. Would it be better to report the geomean */ geostd?

It is the geometric standard deviation (geostd). The first number means the geomean+geostd and the second means geomean-geostd. We changed the explanation in the table as geomean, +geostd, -geostd and reported the three numbers. It should be clear now.

Figure 12: It would be really interesting to use the median size distributions from Fig.5 to compute and overlay lines of constant kappa for each case to evaluate how the box-whiskers fall across the range of hygroscopicities.

We are sorry that it is not totally clear to us what you mean. We assumed you want to see the calculated CCN number concentration by assuming a certain $\kappa$ value, based on the measured median PNSD.

We added the following in page 23, line 2:

"We additionally derived $N_{CCN}$ based on PNSDs. For that, we assumed values for $\kappa$ of 0.1, 0.2, 0.3, 0.4 or 0.5, and calculated the corresponding $d_{crit}$ at different supersaturations. The integrated particle number concentration in the size range larger than $d_{crit}$ were derived from the median PNSDs during dust type2 and marine periods. These particle number concentrations also can be treated as the predicted $N_{CCN}$ at different supersaturations, as shown in solid (dust type2) and dashed (marine type) lines with different color (indicating different $\kappa$) in Fig. 12(a). As expected, the thus derived $N_{CCN}$ were within the measured $N_{CCN}$ range. Comparing the solid and dashed lines, it can be seen that different aerosol types, i.e., different PNSDs, played an important role in $N_{CCN}$ variation. When looking at the results from the different $\kappa$ values, we found the particle chemical composition did not control $N_{CCN}$, especially when the particle number concentration was very low. These colorful solid and dashed lines connected the $\kappa$ and $N_{CCN}$, which can be helpful for $N_{CCN}$ predictions in modeling studies."

The title of Fig. 12 was changed to:

"(a) Boxplot of $N_{CCN}$ as a function of $\kappa$ for marine type (blue) and dust type2 (black). Whiskers show the 10% to 90% percentiles. The predicted $N_{CCN}$ based on median PNSD and different $\kappa$ values (0.1, 0.2, 0.3, 0.4 and 0.5) are shown in solid (during dust type2 period) and dashed lines (during marine period). (b) $\kappa$ as a function of $d_{crit}$ for marine type (blue) and dust type2 (black). Error bars of $d_{crit}$ and $\kappa$ show 1 standard deviation and 1 geometric standard deviation, respectively."

---

## Author Response (AR2)

Dear reviewer,

Thanks for doing this review again.

Concerning the first comment, we would prefer to keep "part 1" and "part 2" in the title, as these are really two parts of the "Characterization of aerosol particles at Cape Verde close to sea and cloud level heights", with part 1 dealing with particle number concentrations, size distributions, CCN concentrations and types of aerosol into different types, while part 2 then adds the information of INP number concentrations. As both manuscripts are about equally long, it would not be reasonable to combine them into one.

Another reason for using the numbering is, that if both manuscripts are published they are both "Gong et al., 2020", and it is, from a readers perspective, much easier to discriminate between " part 1" and" part 2" instead of referring to the more complicated titles.

But we do see the point that numbering the two studies only makes sense if both of the manuscripts will be published. The answers to the reviews of the second manuscript have already been resubmitted and we await the answer, and we suggest to the editor (and the reviewer), that we could wait with a decision on the final title until the outcome on the second work will be clear. In case of a rejection, we'd delete the "part 1", but in case of acceptance, we'd be in favor of keeping it.

Concerning the second comment, as this paper focus on CCN number concentration, it is better to classify particle sources based on particle number instead of particle mass (which is the case when focusing on chemical composition). But in any case, there is no big difference in the characterization done in Fomba et al. (2014) and ours. What is called type A and B in Fomba et al. (2014) is comparable to our marine type and dust type2, respectively. Our dust type 1 is close to type C in Fomba et al. (2014), and our mixture type is close to type D. Type E in Fomba et al. (2014) are the remaining back trajectories that could not be assigned to the above four major classes. Note that no criteria of classification of backward trajectory was explained in Fomba et al. (2014). Fig. 1 in Fomba et al. (2014) only shows one day backward trajectory as an example. This impedes a direct comparison of the two classifications. But we will include this roughly comparison in the new version.

We added the following in page 16, line 10 (new version):

[revised manuscript text omitted]

The dry density of Saharan dust particles was determined in a range of $\rho$ = 2450 - 2700 kg m$^{-3}$ over the Cape Verde Islands (Haywood et al., 2001). The dry particle density of sodium chloride is known to be $\rho$ = 2160 kg m$^{-3}$. The overall effective density of the dust and sea-salt fraction is approximately 2, as recommenced in Schladitz et al. (2011).

The dry dynamic shape factor $\chi$ of mineral dust is $\chi$ = 1.25 (Kaaden et al., 2009) for 1 $\mu$m particles, whereas the dynamic shape factor for sodium chloride is $\chi$ = 1.08 (Kelly and McMurry, 1992; Gysel and Stratmann, 2013). We used the average shape factor of 1.17 in this study.

Based on these, a conversion from aerodynamic to geometric diameters were done for the APS data, and particle number concentrations from the APS were used to correct the multiply charged particle concentrations in the upper size range where the MPSS measured.

[revised manuscript text omitted]

10 Atmos. Chem. Phys., 18, 4477–4496, https://doi.org/10.5194/acp-18-4477-2018, https://www.atmos-chem-phys.net/18/4477/2018/, 2018.

Kaaden, N., Massling, A., Schladitz, A., Müller, T., Kandler, K., Schütz, L., Weinzierl, B., Petzold, A., Tesche, M., Leinert, S., Deutscher, C., Ebert, M., Weinbruch, S., and Wiedensohler, A.: State of mixing, shape factor, number size distribution, and hygroscopic growth of the Saharan anthropogenic and mineral dust aerosol at Tinfou, Morocco, Tellus B, 61, 51–63, https://doi.org/doi:10.1111/j.1600-0889.2008.00388.x, https://onlinelibrary.wiley.com/doi/abs/10.1111/j.1600-0889.2008.00388.x, 2009.

15 Kelly, W. P. and McMurry, P. H.: Measurement of Particle Density by Inertial Classification of Differential Mobility Analyzer–Generated Monodisperse Aerosols, Aerosol Science and Technology, 17, 199–212, https://doi.org/10.1080/02786829208959571, https://doi.org/10.1080/02786829208959571, 1992.

Kristensen, T. B., Müller, T., Kandler, K., Benker, N., Hartmann, M., Prospero, J. M., Wiedensohler, A., and Stratmann, F.: Properties of cloud condensation nuclei (CCN) in the trade wind marine boundary layer of the western North Atlantic, Atmos. Chem. Phys., 16, 2675–2688,

20 https://doi.org/10.5194/acp-16-2675-2016, http://www.atmos-chem-phys.net/16/2675/2016/, 2016.

Schladitz, A., Müller, T., Nowak, A., Kandler, K., Lieke, K., Massling, A., and Wiedensohler, A.: In situ aerosol characterization at Cape Verde, Part1: Particle number size distributions, hygroscopic growth and state of mixing of mrine and Saharan dust aerosol, Tellus B, 63, 531–548, https://doi.org/10.1111/j.1600-0889.2011.00569.x, http://dx.doi.org/10.1111/j.1600-0889.2011.00569.x, 2011.

von der Weiden, S. L., Drewnick, F., and Borrmann, S.: Particle Loss Calculator - a new software tool for the assessment of the performance

25 of aerosol inlet systems, Atmos. Meas. Tech., 2, 479–494, https://doi.org/10.5194/amt-2-479-2009, http://www.atmos-meas-tech.net/2/479/2009/, 2009.